# ADVANCING THE ADVERSARIAL ROBUSTNESS OF NEURAL NETWORKS FROM THE DATA PERSPECTIVE

## ABSTRACT

Robustness in machine learning is a widespread concept and one of the pillars of trustworthiness, ranging from a model's resistance to noise—benign and adversarial—to the reliability of benchmarking. In this work, we analyse the robustness of labelled data which we argue corresponds to the data manifold's curvature as perceived by a model during training. This view can explain the weaker effect of adversarial training in regions of nearby differently labelled data points where a robust decision boundary exists but is not found. In addition, it can explain why a focus on a minimal number of points from these regions during training leads to an increased robust generalisation accuracy beyond the expected local improvements. Combined with minor adjustments to the learning rate, we improve several state-of-the-art results in a cost-neutral way.

## 1 INTRODUCTION

As one of the pillars of trustworthiness in machine learning, robust and reliable decision-making is fundamental for any real-life application scenario. A prominent problem combining data, training and evaluation concerns the adversarial robustness of deep neural networks. These models leverage the knowledge hidden in vast amounts of data, but their vulnerability to adversarial noise (Szegedy et al., 2014) undermines their trustworthiness in practice. Thus, searching for ways to render these otherwise powerful machines more robust to adversaries is imperative.

One prominent defence crafts adversarial noise to augment the data during training which significantly increases robustness (Madry et al., 2018), especially when combined with large amounts of generated data (Wang et al., 2023). To improve the dynamics behind this *adversarial training*, researchers devised several modifications, for example, by adaptively adjusting the attack (Cai et al., 2018; Ding et al., 2018; Zhang et al., 2020a; Wang et al., 2021) or emphasising and re-weighting difficult points (Zeng et al., 2021; Liu et al., 2021), often from regions closest to a model's decision boundary (Zhang et al., 2020b; Xu et al., 2023). Whereas most authors focus on increasing the margin (or distance) between the model's decision boundary and the to-be-classified data points, there exists an interesting phenomenon which margin variation cannot explain entirely: the much weaker effect of adversarial training in regions of nearby differently labelled points.

Naturally, the distance between two differently labelled points limits the margins for any robust decision boundary as exemplified in Fig. 1 (a, top) using examples from the MNIST set (Lecun et al., 1998). The closer such points are the smaller their *natural* margin, leaving less space for a robust decision boundary. In this sense, we can extend the concept of robustness to data points, where the least robust points enforce the narrowest margins. However, while "less space" may explain the weaker effect of adversarial training in such non-robust regions, it fails as an explanation if "less space" is "still more than enough space". In this work, we provide a new perspective on this phenomenon which can explain why these non-robust points are "difficult" for any model and why adversarial training may not lead to a robust decision boundary, even if we know it exists.

Consider Fig. 1 (b) as an example. Both plots display the (adversarial) training and test accuracy on CIFAR-10 (Krizhevsky, 2009) using adversarial training over 2400 epochs with 20 million additional images from Wang et al. (2023). The training attack uses the TRADES loss (Zhang et al., 2019) for a 10-step projected gradient descent (PGD-10) with maximum $l_\infty$ perturbation radius $8/255$ and step size $2/255$ (Madry et al., 2018). By fixing either the 1024 most (left) or least robust images as part of the training set and using a stronger 40-step PGD attack, we establish a learning

focus on these regions. As shown by the red pentagons, the final adversarial accuracy on these sets differs by a factor of 2, with the model remaining much more vulnerable in non-robust regions.

However, smaller margins alone cannot explain this behaviour: the minimal $l_\infty$ distance between two non-robust CIFAR-10 images ($\approx 0.2118$) is more than six times as large as the maximum perturbation radius ($\approx 0.0314$) which, geometrically, leaves more than enough space for a robust decision boundary. Consequently, the distance between differently labelled points cannot explain the weaker performance of adversarial training alone. As our primary contribution in this work, we provide an explanation for this phenomenon by establishing a link between data robustness and the curvature of the data manifold as *perceived by a model during training*.

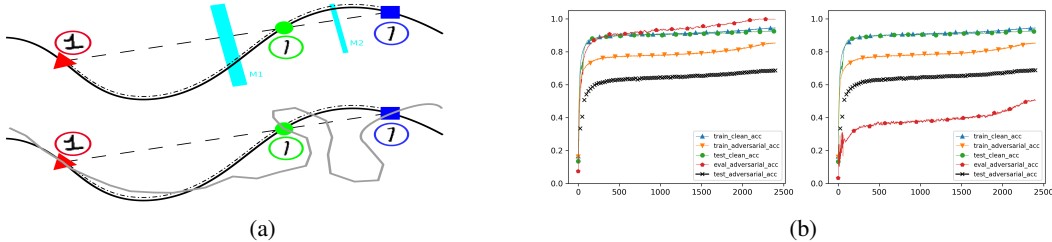

(a)            (b)

Figure 1: (a): Top: data manifold for digits (black curve) together with distances between points using the feature space metric (length of the dashed line) and the manifold metric (length of the dotted+dashed curve). The labels of the triangle, circle and square are "1", "1", and "7", respectively. The bars M1/M2 indicate margins between differently labelled points. Bottom: an example of the learned manifold representation (grey spline). (b): Clean/adversarial training/test accuracy of two models, where the red pentagons display the adversarial accuracy when focusing adversarial training on the 1024 most (left) and least robust elements in the training set.

Let us illustrate what we mean by the perception of curvature: continuing the example in Fig. 1 (a), the lengths of the dashed line and the dotted+dashed line visualise the feature space and manifold distance between points, respectively. While the former measures the "pixel distance", the latter measures the semantic variation of the images. The (discrete) curvature at a data point, e.g. the circle, can be understood as a ratio between these distances to nearby points, effectively comparing the semantic change with the change in terms of pixels. Both distances are similar (e.g., between the circle and square) if and only if the curvature is low. Now, according to Brahma et al. (2016), a network learns its own representation of the data where it disentangles the manifold's structure. However, since classification labels only encode *whether* points are semantically different (in the form of classes) and not the *degree* of these semantic differences, the model may *perceive* the pixel distance as much smaller than the manifold distance in non-robust regions. This can result in a distorted representation of curvature which sustains the susceptibility to adversarial noise as indicated by the bend between the rightmost two images of the learned manifold representation (grey spline) in Fig. 1 (a). More precisely, by "forcing" the model to maximise the distances between differently labelled regions of the data, the training algorithm *also* forces the model to connect pixel distances with the class-changing semantic distances. Since the ratio between these distances is distorted for the least robust points, where the pixel differences need to explain relatively *larger* semantic differences, the model may learn spurious correlations between the primitive features (pixels) and the labels in these regions. As we will show theoretically and empirically, there exists precisely such a link between the robustness of the data and the perceived curvature that can explain the weaker effect of adversarial training in non-robust regions.

To demonstrate that this new geometric understanding is meaningful in practice and can motivate new adversarial training methods, we slightly modify the pipeline of Wang et al. (2023) who provide state-of-the-art adversarial robustness for different datasets and norms. In several experiments, we show that focusing adversarial training on a negligible set of non-robust data points ($\approx 0.005\%$ of the training data) leads to higher robustness for unseen data beyond the expected local improvements. Combined with minor adjustments to the learning rate, we improve the previous state-of-the-art while using *identical computational resources*. To conclude, our contributions are: **(i)** We provide a new and rigorous explanation for the weaker effect of adversarial training in regions of non-robust data by linking the problem to the model's perception of the data manifold's curvature. **(ii)** We reframe the theoretical concept of the data manifold in terms of metric geometry and provide several

results as empirical evidence for our proposed view. **(iii)** We slightly modify the current state-of-the-art pipeline and train more robust models using identical computational resources.

## 2 RELATED WORK

**Adversarial Robustness and Training:** After ten years (Szegedy et al., 2014), the adversarial robustness of neural networks is still an active field of research (Gowal et al., 2020; Rebuffi et al., 2021; Wang et al., 2023). By now, there exist leaderboards (Croce et al., 2021) to track advances for selected image datasets, limitations of which are discussed by Lorenz et al. (2022). Meanwhile, several authors have created adversarial examples in real-world scenarios (Xu et al., 2020; Kurakin et al., 2017; Eykholt et al., 2018), still questioning the trustworthiness of models in practice. Nonetheless, the $l_\infty$ robustness on small-scale datasets such as MNIST (Lecun et al., 1998) and CIFAR-10 (Krizhevsky, 2009) has surpassed the 96% (Gowal et al., 2020) and 70% thresholds (Wang et al., 2023), respectively. Such advances can be attributed to a combination of several factors, including large and wide models (Zagoruyko & Komodakis, 2016), additional data (Carmon et al., 2019; Ho et al., 2020; Karras et al., 2022), weight averaging (Izmailov et al., 2018), label smoothing (Szegedy et al., 2015) and, crucially, adversarial training (Madry et al., 2018) with the TRADES loss function (Zhang et al., 2019). Adversarial training, in particular, exists in many forms: from adaptively adjusting attacks (Cai et al., 2018; Ding et al., 2018; Zhang et al., 2020a; Wang et al., 2021) to emphasising and re-weighting difficult points (Zeng et al., 2021; Liu et al., 2021), often from regions closest to a model's decision boundary (Zhang et al., 2020b; Xu et al., 2023).

**Diffusion Models:** As cheap data providers, diffusion models have attracted much attention recently (Ho et al., 2020; Karras et al., 2022; Rombach et al., 2022), especially because of their improvements over generative adversarial networks (Dhariwal & Nichol, 2021). In essence, diffusion models learn to reverse the process of turning signal into noise by predicting noise residuals (for example, on images), enabling them to generate new data points from pure noise (Sohl-Dickstein et al., 2015). Once trained, they can provide endless semantically-different examples to augment the training data. On the other hand, diffusion models can also improve the adversarial robustness of models when included as input "purifiers", effectively removing the adversarial noise (Nie et al., 2022).

**Data Manifold:** Assuming that the data manifold is embedded in Euclidean space, it seems to be of drastically smaller dimension than the ambient space, especially for image data (Pope et al., 2021; Brown et al., 2022). Apart from artificial examples (Cayton, 2005; Brahma et al., 2016), proving its existence is complex (Fefferman et al., 2013). Nonetheless, information about this latent structure benefits a neural network's performance via unsupervised pre-training (Erhan et al., 2010), where the disentangled manifold appears to be flattened in the higher layers (Brahma et al., 2016). In this work, we use concepts from metric geometry (Rinow, 1961; Gluck, 1966; Gromov, 2007; Bridson & Häfliger, 2011) to re-frame the data manifold and employ a specific formulation of (arc) curvature (Kay, 1980; Saucan, 2017), initially introduced by Haantjes (1947).

## 3 BACKGROUND AND THEORETICAL FRAMEWORK

We begin by regarding the data as decoupled from the assigned labels and the data manifold as *arc-connected* as long as the data features are continuously distributed. For images, Brown et al. (2022) claim that there exist several class-dependent, disconnected manifolds of varying intrinsic dimensionality, arguing that an image of a "2" cannot be continuously transformed into an image of an "8". In contrast, we assume that such a transformation exists along a curve tangent to the data manifold (comp. Fig. 1 (a)) for which well-defined intermediate labels merely do not exist. *In other words, the data manifold exists independent of any labels.* To give an intuitive example, one may imagine shooting a film, starting with an image of a "2" before exchanging it for an image of an "8". Naturally, this provides a continuous stream of images transforming the former into the latter, implying that the data manifold for such images is arc-connected and, a fortiori, connected.

As is customary, we regard the (image) data as elements in the hypercube $[0, 1]^n$, where $n$ is the number of features. Because multiple metrics may be interesting (such as the Euclidean ($l_2$) and infinity ($l_\infty$) norm for images), it makes sense to work with a (compact) metric space $(X, d)$, for example, $([0, 1]^n, l_{2_{|[0,1]^n}})$. Given $C \in \mathbb{N}$ classification labels, we assume that the components of the corresponding one-hot-encoded labels represent the values of $C$ real-valued probability densities.

More precisely, we define a *label map* $y : X \to [0,1]^C$ as a continuous extension of the one-hot-encoded labels $y_i := y(p_i)$ from a finite dataset $\mathcal{D} := \{p_1, \ldots, p_m\} \subset X$.

To construct a suitable notion of the data manifold, a few preliminary steps are necessary (Rinow, 1961; Gluck, 1966; Gromov, 2007; Bridson & Häfliger, 2011): first, assume we have some arc-connected set $M \subset X$ such that the arc between any two points is rectifiable, that is, of finite length. The length of an arc $\gamma : [0,1] \to M$ connecting two points $p_1, p_2 \in M$ is defined as $L(\gamma) := \sup\{l(R) \ : \ R = \{0 = r_0 \leq r_1 \leq \cdots \leq r_{k_R} = 1\}, k_R \in \mathbb{N}\}$ with $l(R) := \sum_{i=1}^{k_R} d_{|M}(\gamma(r_{i-1}), \gamma(r_i))$. One can show that the length functional in this case induces a well-defined metric $\tilde{d}$ on $M$ via $\tilde{d}(p_1, p_2) := \inf_{\gamma \in \Gamma(p_1,p_2)} L(\gamma)$, where $\Gamma(p_1, p_2)$ is the set of rectifiable arcs in $M$ connecting $p_1$ and $p_2$. It follows that $\tilde{d} \geq d_{|M \times M}$ as real-valued functions.

Finally, the metric space $(M, \tilde{d})$ is what we regard as the data manifold, where $\tilde{d}$ is the manifold metric measuring semantic variation along (or constrained to) the manifold (comp. Fig. 1 (a)). Note that this construction circumvents the problem of assuming a differentiable structure on the manifold or the latter being of constant intrinsic dimension which would contradict findings by Brown et al. (2022) building on results from Pope et al. (2021). Note furthermore that the continuity of $y$ on $X$ w.r.t. $d$ implies the continuity of $y$ on $M$ w.r.t. $\tilde{d}$, see (Gluck, 1966). Fig. 1 (a) illustrates such a data manifold and the distance in terms of the length of curves connecting two points.

A rectifiable arc $\gamma$ has Finsler-Haantjes curvature (Haantjes, 1947; Kay, 1980; Saucan, 2017) $\kappa(p)$ at $p \in M$ if $\kappa(p_1, p_2) \to const. =: \kappa(p) < \infty$ for $p_1, p_2 \to p$, where $\kappa(p_1, p_2) := \left(4! \frac{\tilde{d}(p_1,p_2)-d(p_1,p_2)}{d(p_1,p_2)^3}\right)^{\frac{1}{2}}$. One can show that this generalises the classical curvature for smooth arcs in Euclidean vector spaces (Kay, 1980). An equivalent (and more convenient) expression is:

$$\tilde{\kappa}(p_1, p_2) := \frac{\kappa(p_1,p_2)^2}{4!} = \frac{\tilde{d}(p_1,p_2)-d(p_1,p_2)}{d(p_1,p_2)^3}$$

The more $\tilde{d}$ and $d$ diverge for curve points $p_1, p_2$ converging to some $p$, the larger the curvature at $p$. For images, a large curvature in this sense can be visible as a strong semantic change (w.r.t. $\tilde{d}$) relative to the pixel difference (w.r.t. $d$). Unfortunately, $\tilde{d}$ is inaccessible, so it will not allow us to draw any conclusions about the actual curvature. However, as stated in the introduction, we are interested in the curvature of the data manifold *as perceived by a model during training*. The following functional will provide the means to test this in practice:

$$s_y : \mathcal{D} \to \mathbb{R}, \quad p_i \mapsto \max_{p_i \neq p_j \in \mathcal{D}} \frac{\|y(p_i)-y(p_j)\|}{d(p_i,p_j)} \quad (1)$$

This expression resembles a "point-wise" Lipschitz constant of the label map $y$ on the finite dataset $\mathcal{D}$ and, thus, we will call $s_y(p_i)$ the *sensitivity of $p_i$*. To connect this formulation to the concept of robustness outlined in the introduction, we can simply define $r(x_i) := \min_{x_j : y(x_i) \neq y(x_j)} d(x_i, x_j)$ as the robustness of a data point. Both concepts are related and provide equivalent information if $y$ takes only binary values. In this sense, the least and most sensitive elements are the most and least robust, respectively. If we assume $y$ to be locally constant around each $p_i \in \mathcal{D}$, we have the following result revealing the measure's local character:

**Theorem 1.** Let $(X, d)$ and $(Y, \|\cdot\|)$ be a metric and Banach space, respectively, and $\{p_1, \ldots, p_m\} =: \mathcal{D}$ a finite collection of points. Assume that there exists a map $y : X \to Y$ which is locally constant around each $p_i$. Then there exists an open set $U_i \subset X$ around any fixed $p_i$ such that the following functional is Lipschitz continuous:

$$s_y : U_i \to \mathbb{R}, \ x \mapsto \max_{p_i \neq p_j \in \mathcal{D}} \frac{\|y(p_j)-y(x)\|}{d(p_j,x)}$$

We present a proof in Appendix A. Since $|s_y(a)-s_y(b)| \leq L_i d(a,b)$ for some $L_i := L(U_i) \geq 0$ and $a, b \in U_i$, Theorem 1 guarantees that sensitivity information is locally consistent (both for $(X, d)$ *and* $(M, \tilde{d})$) as long as the same holds for the label map. In particular, the information provided by $s_y$ is robust under small enough (adversarial) noise, label smoothing (Szegedy et al., 2015) with a uniform distribution and rescaling of $y$ (see Appendix A, Remark 1).

We now derive our main assumption from empirical evidence. Since the features and labels are all the information conveyed to a model during training, the manifold distance—as perceived by the

model—can, at most, incorporate the same information. To make this thought explicit, we express the perceived manifold distance as $\tilde{d}_{perc}(y, p_i, p_j)$. Now, given sufficient capacity, neural networks can learn any function on a reasonably well-behaved space (Hornik, 1991), but—in terms of training optimisation—it seems irrelevant what the signal-to-noise ratio is or whether labels are assigned at random (Zhang et al., 2016; 2021). Although correct labels allow faster training (Frankle et al., 2020), training with random labels transfers knowledge to the model (Maennel et al., 2020) which can not entirely be explained by memorisation (Arpit et al., 2017; Pondenkandath et al., 2018).

Based on these empirical results, it appears that the particular choice of $y$ and, thus, the perceived manifold distance between differently labelled elements $\tilde{d}_{perc}(y, p_i, p_j)$ for $y(p_i) \neq y(p_j)$ do not have any (substantial) impact on training optimisation. We, therefore, argue that a model perceives $\tilde{d}_{perc}(y, p_i, p_j) = \delta(y)$ for some constant $\delta(y) > 0$ if and only if $p_i$ and $p_j$ have distinct labels and 0 otherwise (which is exactly the information contained in $\|y(p_i) - y(p_j)\|$ for one-hot encoded $y$). Up to rescaling of the norm, we write: $\|y(p_i) - y(p_j)\| = \tilde{d}_{perc}(y, p_i, p_j)$. Since the three sets $\Delta_1 := \{d(p_i, p_j)^{-1}\}$, $\Delta_2 := \{\lambda - d(p_i, p_j)\}$ and $\Delta_3 := \{d(p_i, p_j)^{-3}\}$ admit the same order for any $\lambda \in \mathbb{R}$, where $p_i, p_j \in \mathcal{D}, y(p_i) \neq y(p_j)$, the same holds for $\Delta_4 := \{\frac{\lambda - d(p_i, p_j)}{d(p_i, p_j)^3}\}$. Exchanging $\tilde{d}$ for $\tilde{d}_{perc}$ to turn $\tilde{\kappa}$ and $\kappa$ into $\tilde{\kappa}_{perc}$ and $\kappa_{perc}$, respectively, we can write $\Delta_4 = \{\tilde{\kappa}_{perc}(p_i, p_j)\}$ when setting $\lambda := \delta(y)$. Finally, because $\Delta_1$ and $\Delta_4$ have the same order of elements and $\tilde{\kappa}_{perc}$ is proportional to $\kappa_{perc}$, the sets $\{s_y(p_i) \mid p_i \in \mathcal{D}\}$ and $\{\max_{p_j:y(p_i)\neq y(p_j)} \kappa_{perc}(p_i, p_j) \mid p_i \in \mathcal{D}\}$ have the same order as well.

If $p_i \in \mathcal{D}$, let $\gamma_j$ be an arc in $M$ connecting $p_i = \gamma_j(0)$ to $p_j = \gamma_j(1)$ for $y(p_j) \neq y(p_i)$. Then $\max_{p_j:y(p_i)\neq y(p_j)} \kappa_{perc}(p_i, p_j) = \max_{\gamma_j} \kappa_{perc}(\gamma_j(0), \gamma_j(1))$ approximates the maximum perceived curvature along curves starting at $p_i$ and ending in regions of points with different labels. We now see that, under our assumption that a model perceives differently labelled elements as having the same manifold distance $\delta(y)$, it also perceives the most and least sensitive elements of the data manifold as the elements with the highest and lowest curvature, respectively. This may explain why models struggle more with adversarial noise in the most sensitive / least robust regions, see Fig. 1 (b): as the ratio between $\tilde{d}$ and $d$ is distorted for the least robust points—where the pixel differences need to explain relatively *larger* semantic differences—the model may learn spurious correlations between the primitive features (pixels) and the labels in these regions. Interestingly, Arpit et al. (2017) have shown that adversarial examples are easier to generate when training a network with random labels. Since random labels can increase the sensitivity values of data points, comp. Equation (1), this aligns perfectly with our view. Finally, from Theorem 1 it follows that any $q \in M$ with $d(p_i, q)$ small enough will have a similar sensitivity value $s_y(p_i) \approx s_y(q)$ and, repeating the above arguments, a similar approximate curvature $\max_{p_j:y(p_i)\neq y(p_j)} \kappa_{perc}(p_i, p_j) \approx \max_{p_j:y(q)\neq y(p_j)} \kappa_{perc}(q, p_j)$. In this sense, data sensitivity/robustness and *perceived* curvature are local phenomena.

## 4 EVALUATION

In this section, we will **(i)** motivate empirically why data sensitivity/robustness and perceived curvature are related as explained in Section 3 (Figure 2 (a, b)); **(ii)** provide visual evidence for Theorem 1 (Figure 2 (c, d)); **(iii)** showcase the link between data robustness and a (diffusion) model's perception of the data manifold's curvature (Figure 3 & Table 1); **(iv)** show that stronger adversarial training on a negligible number of non-robust points ($\approx 0.005\%$) entails robustness benefits beyond the expected local improvements (Table 2) and **(v)** improve the state-of-the-art approach from Wang et al. (2023) using identical computational resources (Tables 3 and 4).

**Datasets:** For the data analysis, we used the CIFAR-10/100/100s training sets (Krizhevsky, 2009), together with the 500,000 pseudo-labelled real ("TI500K") and the 1 million generated ("1m") 10-class datasets from Carmon et al. (2019) and Wang et al. (2023), respectively. "TI500K" and the CIFAR data are subsets of the 80 Million Tiny Images dataset (Torralba et al., 2008). CIFAR-100s is CIFAR-100 with labels grouped into 20 more general "super" classes. For our adversarial robustness experiments, we use CIFAR-10/100. We focus on the CIFAR data because training pipelines and diffusion models are available or quickly adopted. For CIFAR-100, we excluded a small set of 40 duplicate images. Appendix C displays results for the EMNIST set (Cohen et al., 2017).

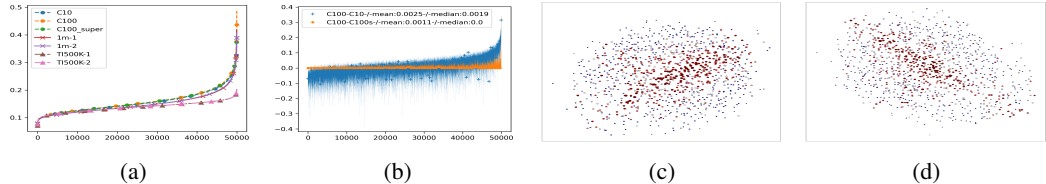

(a)             (b)             (c)             (d)

Figure 2: (a): Ordered $l_2$ sensitivity values for the original CIFAR data and subsets of "TI500K" and "1m". (b): The differences between sensitivity values of CIFAR-100/10 ("+") and CIFAR-100/100s ("o"), ordered by the CIFAR-100 values. Right half: *MDS* representations of a CIFAR-10 (c) and CIFAR-100 (d) sample, coloured and resized in proportion to their $l_2$ sensitivity values, where the most and least robust elements are the small blue and large red points, respectively.

**Experimental Setup:** All sensitivity values were calculated according to the formula in Equation (1) w.r.t. the Euclidean ($l_2$) and the infinity ($l_\infty$) norm; see Algorithm 1 and the benchmarks in Appendix B. We used multidimensional scaling (*MDS*) (Kruskal, 1964) which visualises similarities based on pair-wise differences of data points to see whether detectable patterns emerged for a robustness-stratified sample of 1000 elements. The diffusion models are taken from Karras et al. (2022) and Wang et al. (2023) for CIFAR-10 and CIFAR-100, respectively, where we changed the backward diffusion process to mimic a data cloning pipeline. Instead of starting with pure noise, we used a normalised weighted combination of random normal noise (45%) and the original images (55%) to reduce the signal-to-noise ratio without sacrificing the entire information. To optimise the denoising process, we ran a small hyperparameter search over three random seeds to find the minimal FID (Heusel et al., 2017) between the least robust 10,000 CIFAR-10 images (w.r.t. the $l_\infty$ norm) and their clones. The list of parameters is in Appendix B; we refer to the work of Karras et al. (2022) for their precise definition. The final FIDs were 5.83/5.76/5.78. Because of the high computational cost, we used the same parameters for CIFAR-100.

Our evaluation setup for the adversarial robustness experiments matches the state-of-the-art approach by Wang et al. (2023) for the more economical Wide-Res-Net-28-10 architecture (Zagoruyko & Komodakis, 2016) (more details in Appendix B). We used PGD-10-based adversarial training with TRADES loss Zhang et al. (2019) on the CIFAR-10/100 sets for different batch sizes (BS), epochs (E), norms (N) and amounts of additional generated images (GI) from Wang et al. (2023). The adversarial robustness was evaluated on the test sets using AutoAttack (Croce & Hein, 2020). During our experiments, we uncovered a glitch in the training pipeline of Wang et al. (2023), which we used to include stronger adversarial noise in the training process. We describe the dynamics in Appendix D. However, we were able to circumvent the glitch and emulate its effect, as discussed below.

**The (Local) Consistency of Sensitivity Values:** We show a selection; more results are in Appendix B. Figure 2 (a) displays the ordered $l_2$ sensitivity values for CIFAR-10/100/100s and two random samples of 50,000 elements from the "TI500K" and "1m" datasets. Figure 2 (b) shows the differences between sensitivity values of CIFAR-100/10 ("+") and CIFAR-100/100s ("o"), ordered by the CIFAR-100 values. Compared to the "TI500K" data, all other sets have similar sensitivity distributions. Whereas the slight differences for the "1m" subsets can be explained by using a classifier's predictions as labels (similar as for "TI500K"), the more noticeable differences for "TI500K" can be explained by Carmon et al. (2019) removing near-duplicates of the CIFAR-10 test set. More precisely, as sensitivity is a local concept (proven in Theorem 1), removing near-duplicates from a set can decrease the values independent of the reference set. Both graphs indicate that the sensitivity values for distinct datasets from the same manifold depend primarily on feature information and not on the granularity of the class distributions. As we have outlined in Section 3, this aligns sensitivity information with how a model perceives curvature. Figure 2 (c) and (d) display the *MDS* representations for CIFAR-10 and CIFAR-100, respectively. Patterns emerge based on the point colours and sizes, showcasing the statement of Theorem 1: similar points—which will be clustered together by the algorithm—have similar sensitivity values. We note that *MDS* does not use label information.

**A Model's Perception of Curvature:** Figure 3 displays the original 16=4x4 most and least robust elements (w.r.t. the $l_2$ norm) from CIFAR-10/100 with their clones, respectively. More detailed examples are shown in Figures 8-15 in Appendix B. The most robust elements have semantically similar clones compared to the least robust elements. The differences for the former mainly consist

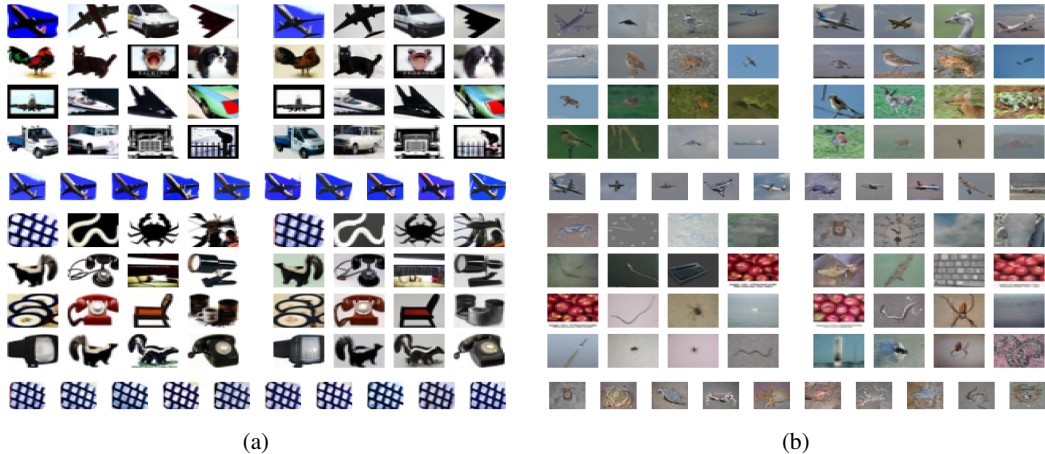

|                                        (a)                                       |                                        (b)                                       |

Figure 3: (a) Square arrangements of the 16=4x4 most robust images for CIFAR-10 (top left) and CIFAR-100 (bottom left) together with their individual clones on the right and collections of ten clones for the first image, respectively. (b) The corresponding least robust images and clones.

Table 1: Pixel distances and FID between the 10,000 least/most robust images and their clones (O:C) and the pixel distances between 10 clones of the 1024 least/most robust images (C:C), respectively.

| Set | $\|\cdot\|$-Rob. | $l_\infty$ (O:C) | $l_2$ (O:C) | FID (O:C) | $l_\infty$ (C:C) | $l_2$ (C:C) |
|---|---|---|---|---|---|---|
| CIFAR-10 | $l_2$-Most | 0.63 | 7.73 | 5.54 | 0.72 | 8.99 |
| CIFAR-10 | $l_2$-Least | 0.53 | 6.89 | 5.59 | 0.55 | 6.99 |
| CIFAR-10 | $l_\infty$-Most | 0.63 | 7.61 | 5.64 | 0.70 | 8.75 |
| CIFAR-10 | $l_\infty$-Least | 0.52 | 7.15 | 5.76 | 0.55 | 7.44 |

of colour changes, either of the object or background; conversely, the semantic changes for the least robust elements are more pronounced and include the object's general shape and surroundings. This is (visual) evidence of the diffusion model connecting more significant semantic differences to regions of the least robust elements.

Table 1 shows the average $l_2$ and $l_\infty$ distances, and the FID between the 10,000 least/most robust CIFAR-10 elements (w.r.t. both norms) and their clones (columns 3,4,5). The last two columns show the corresponding average distances between 10 clones of the same image for the 1024 most and least robust elements, respectively (we measured the distance between each clone and its follow-up clone). The statistics for CIFAR-100 are similar (see Table 5 in Appendix B). Interestingly, the average "pixel distances" (w.r.t. the $l_2$ and $l_\infty$ norm) are *smaller* for the least robust elements than for the most robust when comparing original images with clones (O:C) and when comparing clones with clones (C:C), respectively. Based on these differences between the pixel distance and the semantic manifold distance (or $d$ and $\tilde{d}$ as introduced in Section 3), the diffusion models seem to produce fundamentally different clones for the least (large $\tilde{d}$, small $d$) and most (small $\tilde{d}$, large $d$) robust images. In terms of the Finsler-Haantjes curvature $\kappa$, the model connects the least and most robust elements to regions of high and low curvature, respectively, when generating images.

To provide evidence for our claim that a weaker performance of adversarial training may be related to a model perceiving higher curvature, we tested how a focus on a small set of non-robust points can influence the robustness on unseen data. If there is no such connection, we expect no increase in robust generalisation accuracy beyond the local improvements on the small set, indicating that the model's learned representation of the data is largely unchanged. To this end, we slightly modified the adversarial training on CIFAR-10 with N=$l_\infty$/E=2400/BS=2048/GI=20m. Table 2 displays the results, where $v$ and $v'$ denote distinct random sets with 1024 elements, while $s$ contains the 1024 least robust elements; the label distributions are displayed in Fig. 4 (a) and (b). The minus sign "-" indicates which set was held out from the training data, whereas the set in parentheses was *focused* on during training. This focus comes in the form of a stronger PGD-40 attack in every training epoch. Importantly, if the set in parentheses is not simultaneously held out, it is part of the training data which allows us to emphasise learning on its elements.

Table 2: Results for CIFAR-10 with $l_\infty$ norm ($C10_\infty$). $AA_b$ and $AA_l$ display the clean (left value) and adversarial accuracy for the "best" and "last" weights in columns 3 and 4, respectively, with corresponding deltas in columns 5 and 6. Here, the "best" weights are determined by the epoch $E_b$ (column 7) where the best PGD-40 accuracy (column 8) on the set in parentheses was achieved.

| Description | Setup | $AA_b$ | $AA_l$ | $\Delta_b$ | $\Delta_l$ | $E_b$ | $(PGD\text{-}40)_b$ |
|---|---|---|---|---|---|---|---|
| Wang et al. (2023) | $C10_{\infty,-v}(v)$ | 92.45 - 67.53 | 92.40 - 67.57 | 24.92 | 24.83 | 2384 | 72.66 |
| Validation set (VS) | $C10_{\infty,-v'}(v)$ | 92.62 - 67.54 | 92.63 - 67.56 | 25.08 | 25.07 | 2393 | 89.75 |
| | $C10_{\infty,-v}(\cdot)$ | x | 92.47 - 67.52 | x | 24.95 | x | x |
| | $C10_{\infty,-s}(s)$ | 92.44 - 67.57 | 92.54 - 67.47 | 24.87 | 25.07 | 2340 | 39.94 |
| | $C10_{\infty,-v}(s)$ | 92.40 - **67.77** | 92.40 - **67.70** | **24.63** | **24.70** | 2393 | 50.98 |
| Reproducibility | $C10_{\infty,-v}(2s)$ | 92.59 - **67.72** | 92.56 - **67.72** | 24.87 | 24.84 | 2390 | 55.62 |
| | $C10_{\infty,-v}(4s)$ | 92.45 - **67.72** | 92.42 - 67.71 | **24.73** | **24.71** | 2396 | 62.65 |
| Learning rate (LR) | $C10_{\infty,-v,a}(s)$ | 92.60 - **67.61** | 92.46 - 67.72 | 24.99 | 24.74 | 2382 | 52.25 |
| | $C10_{\infty,-v,b}(s)$ | 92.55 - 67.55 | 92.59 - 67.55 | 25.00 | 25.04 | 2383 | 51.46 |
| | $C10_{\infty,-v,c}(s)$ | 92.44 - 67.58 | 92.47 - **67.83** | **24.86** | **24.64** | 2238 | 51.17 |
| | $C10_{\infty,-v,d}(s)$ | 92.41 - 67.41 | 92.41 - 67.47 | 25.00 | 24.94 | 2293 | 50.88 |
| | $C10_{\infty,-v,e}(s)$ | 92.57 - 67.51 | 92.51 - 67.45 | 25.06 | 25.06 | 2386 | 51.86 |

**Training Focus and Robust Generalisation:** As expected, we notice a consistent increase in PGD-40 accuracy (last column) for the sets in focus when simultaneously including them in the training set: $\approx 17$ and $\approx 11\%$ percentage points for $v$ (comp. rows 1&2) and $s$ (comp. rows 4&5), respectively. The first row shows the reproduced performance of Wang et al. (2023) who use $v$ as a hold-out set; rows 2 and 3 show variations to estimate the effect of including stronger adversarial noise for randomly selected data ("$(\cdot)$" indicates that no set was used). The adversarial robustness on the test set is consistent (columns 3&4). Rows 4&5 show the results when exchanging $v$ for $s$. We see an increase in adversarial robustness (columns 3&4) by about 0.2 percentage points when including $s$ (row 5). This difference appears small but amounts to the same boost gained by extending training for 800 epochs using identical computational resources (based on results from Wang et al. (2023)).

Even though the stronger adversarial noise concentrates on a negligible number of points compared to the entire training data ($\approx 0.005\%$), the absolute improvement in adversarial accuracy on $s$ ($\approx 112$ images) almost doubles for the test data ($\approx 200$ images) which, in addition, undergo a much stronger ensemble of attacks (Croce & Hein, 2020). Surprisingly, focusing on a negligible number of points from non-robust regions leads to an increased robust generalisation accuracy beyond the expected local improvements. In other words, a better understanding of the pixel distances between points is paired with a better understanding of semantic distances (hence the improved performance on unseen data). Together, the focus may have entailed a better understanding of curvature. However, we also notice that doubling ($2s$) or quadrupling ($4s$) the least robust elements does not entail further benefits, conveying a special significance to the 1024 least robust elements.

To see whether the effect can be improved, we devised minor learning rate adjustments based on the behaviour of the PGD-40 robustness on $s$ which increases significantly for smaller learning rates (comp. Fig. 1 (c) and (d)). A connection between adversarial robustness and the later training stages has also been observed by Wang et al. (2021). We modified the annealing strategy to let learning rates fall off quicker and extended their tail towards the end, see Figure 4 (c, d). Version $a$ and $b$ follow a constant value of $2 \cdot 10^{-4}$ for the last 240 and 400 epochs, version $c$ and $d$ have exponential fall-offs after 2000 and 1900 epochs, ending on $5 \cdot 10^{-5}$ and $1.5 \cdot 10^{-5}$, respectively. Version $e$ combines the exponential fall-off after 2000 epochs with ending on the higher value $2 \cdot 10^{-4}$. Rows 8-12 in Table 2 show the results (again focusing on $s$), where version $c$ led to an improved performance after training and a better adversarial delta overall. The worse performance for the "best" weights may be explained by the latter being achieved too early (epoch 2238).

After this exploratory phase, we re-evaluated the experiments for different dataset/norm combinations and random seeds to see whether the previous results were generalisable. Table 3 displays that this is indeed the case. Here, columns 1-2, 3-4 and 5-6 show performances for CIFAR-10 with N=$l_\infty$/E=2400/BS=2048/GI=20m, CIFAR-10 with N=$l_2$/E=1600/BS=2048/GI=50m and CIFAR-100 with N=$l_\infty$/E=1600/BS=2048/GI=50m, respectively (we modified the previous learning rate version $c$ to match the settings with fewer epochs). We then re-evaluated our experiments once more, see Table 4, by increasing the number of epochs (2400→3000, 1600→2000; again adjusting the learning rate version $c$), the batch size (2048→2080) and doubling the number of images used

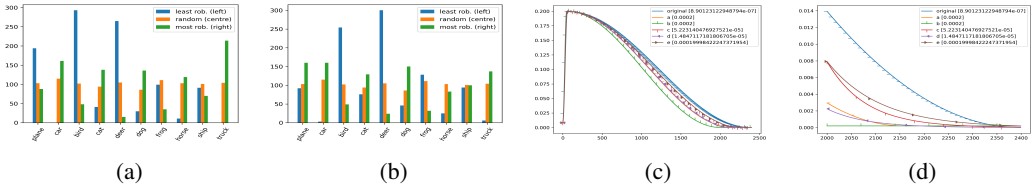

(a)            (b)            (c)            (d)

Figure 4: (a): Label distributions of the least robust (left bars), random (centre bars) and most robust (right bars) 1024 elements with respect to the $l_2$ norm. (b) Analogous distributions for the $l_\infty$ norm. (c): The original and five modified learning rates over 2400 epochs with the final learning rate values in the top right. (d): The last 400 epochs in more detail.

Table 3: Benchmarks with new random seeds for CIFAR-10 with $l_\infty$ ($C10_\infty$, rows 1,2) and $l_2$ norm ($C10_2$, rows 3,4), and CIFAR-100 with $l_\infty$ norm ($C100_\infty$, rows 5,6). Notation as in Table 2.

| Description | Setup | $AA_b$ | $AA_l$ | $\Delta_b$ | $\Delta_l$ | $E_b$ | (PGD-40)$_b$ |
|---|---|---|---|---|---|---|---|
| Wang et al. (2023) | $C10_{\infty,-v}(v)$ | 92.60 - 67.61 | 92.61 - 67.65 | 24.99 | 24.96 | 2377 | 72.56 |
| VS & LR | $C10_{\infty,-v,c}(s)$ | 92.53 - **67.83** | 92.54 - **67.78** | **24.70** | **24.76** | 2319 | 50.98 |
| Wang et al. (2023) | $C10_{2,-v}(v)$ | 95.15 - 83.54 | 95.19 - 83.59 | 11.61 | 11.60 | 1565 | 88.18 |
| VS & LR | $C10_{2,-v,c}(s)$ | 95.10 - **83.61** | 95.11 - **83.69** | **11.49** | **11.42** | 1568 | 84.08 |
| Wang et al. (2023) | $C100_{\infty,-v}(v)$ | 72.97 - 38.24 | 72.80 - 38.39 | 34.73 | 34.41 | 1531 | 46.68 |
| VS & LR | $C100_{\infty,-v,c}(s)$ | 72.78 - **38.54** | 72.78 - **38.58** | **34.42** | **34.20** | 1568 | 32.52 |

Table 4: Extension results. Notation as in Tables 2 and 3, random seeds as in Table 3.

| Description | Setup | $AA_b$ | $AA_l$ | $\Delta_b$ | $\Delta_l$ | $E_b$ | (PGD-40)$_b$ |
|---|---|---|---|---|---|---|---|
| Wang et al. (2023) & VS & E & BS | $C10_\infty(2v)$ | 92.51 - **68.07** | 92.56 - **68.04** | **24.44** | **24.52** | 2987 | 91.02 |
| VS & LR & E & BS | $C10_{\infty,c}(2s)$ | 92.55 - 67.72 | 92.52 - 67.65 | 24.83 | 24.87 | 2999 | 57.67 |
| VS & LR & E & BS & * | $C10_{\infty,c}(2s)$ | 92.50 - 67.79 | 92.52 - 67.78 | 24.71 | 24.74 | 2999 | 88.87 |
| Wang et al. (2023) & VS & E & BS | $C10_2(2v)$ | 95.20 - 83.72 | 95.20 - 83.72 | 11.48 | 11.48 | 2000 | 98.97 |
| VS & LR & E & BS | $C10_{2,c}(2s)$ | 95.23 - **83.80** | 95.26 - **83.88** | 11.43 | 11.38 | 1976 | 89.36 |
| VS & LR & E & BS & * | $C10_{2,c}(2s)$ | 94.99 - **83.80** | 95.05 - 83.80 | **11.19** | **11.25** | 1994 | 99.80 |
| Wang et al. (2023) & VS & E & BS | $C100_\infty(2v)$ | 73.18 - 38.81 | 73.11 - 38.86 | 34.37 | 34.25 | 1989 | 76.12 |
| VS & LR & E & BS | $C100_{\infty,c}(2s)$ | 73.12 - **38.94** | 73.12 - **38.92** | 34.18 | 34.20 | 1924 | 36.33 |
| VS & LR & E & BS & * | $C100_{\infty,c}(2s)$ | 72.91 - 38.91 | 72.93 - 38.91 | **34.00** | **34.02** | 1999 | 79.35 |

with the stronger PGD-40 attack (v→2v, s→2s). In addition, no images were held out. We also mimicked the assumed dynamics of the glitch (see Appendix D) in the runs indicated via "*" by including the stronger adversarial noise batch-wise at a rate of 32=2080-2048. Note that this entirely circumvented the glitch while explaining the larger PGD-40 robustness (last column). Except for $C10_\infty$, we again notice benefits. The worse performances for $C10_\infty$ may indicate overfitting on $2s$ (note that the performance is worse than before), displaying the limitations of this approach.

## 5    CONCLUSION AND FUTURE WORK

In this work, we provide a new and rigorous explanation for the weaker effect of adversarial training in regions of non-robust data by linking the problem to the model's perception of the data manifold's curvature. We reframed the concept of a data manifold and generated elements specifically from robust and non-robust regions to showcase this curvature perception. We then demonstrated that emphasising non-robust regions during adversarial training could lead to higher robustness for unseen data beyond the expected local improvements. Minor adjustments to the training pipeline improved several state-of-the-art results while using identical computational resources. In the future, additional empirical and theoretical evidence may further enlighten the dynamics of a model perceiving curvature to motivate new methods of adversarial training.

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

## APPENDIX A (PROOFS)

The proof of Theorem 1 is straight-forward but requires a few preliminary results (Cobzaş et al., 2019) which we state first:

**Proposition A.** *Let $(X, d)$, $(X', d')$ and $(X'', d'')$ be metric spaces. If $f : X \to X'$ and $g : X' \to X''$ are (locally) Lipschitz continuous, then $g \circ f : X \to X''$ is (locally) Lipschitz continuous. Moreover:*

$$L(g \circ f) \leq L(g) \cdot L(f)$$

**Proposition B.** *Let $(X, d)$ be a metric space, $\mathbb{R}$ endowed with the usual Euclidean metric and $f : X \to \mathbb{R}$ a Lipschitz function having the property that there exists $m > 0$ such that $|f(x)| \geq m$ for all $x \in X$. Then $f^{-1}$ is Lipschitz and:*

$$L(\frac{1}{f}) \leq \frac{L(f)}{m^2}$$

**Proposition C.** *Let $(X, d)$ be a metric space. If the functions $f, g : M \to \mathbb{R}$ are bounded and Lipschitz, then $fg$ is Lipschitz. Moreover:*

$$L(fg) \leq \sup_{x \in X} |f(x)| \cdot L(g) + \sup_{x \in X} |g(x)| \cdot L(f)$$

**Proposition D.** *Let $(X, d)$ be a metric space and $\mathcal{F}$ a family of real-valued $K-$Lipschitz functions defined on $X$ such that $\varphi(x) := \sup\{f(x) \mid f \in \mathcal{F}\}$ is finite for every $x \in X$. Then the function $\varphi$ is $K-$Lipschitz. In particular, for two Lipschitz functions $f, g : X \to \mathbb{R}$, their maximum $\max\{f, g\}$ is Lipschitz with $L(\max\{f, g\}) = \max\{L(f), L(g)\}$.*

**Theorem 1.** Let $(X, d)$ and $(Y, \|\cdot\|)$ be a metric and Banach space, respectively, and $\{p_1, \ldots, p_m\} =: \mathcal{D}$ a finite collection of points. Assume that there exists a map $y : X \to Y$ which is locally constant around each $p_i$. Then there exists an open set $U_i \subset X$ around any fixed $p_i$ such that the following functional is Lipschitz continuous:

$$s_y : U_i \to \mathbb{R}, \ x \mapsto \max_{p_i \neq p_j \in \mathcal{D}} \frac{\|y(p_j) - y(x)\|}{d(p_j, x)}$$

**Proof:** There exists an open set $U_i \subset X$ around $p_i$ on which $y$ is constant with $U_i \cap \{p_1, \ldots, p_m\} = \{p_i\}$ and $p_j \notin \partial U_i \ \forall j$. Naturally, the maps $x \mapsto d(x, p_j)$ and $x \mapsto \|y(x) - y(p_j)\|$ are Lipschitz on $U_i$ for all $j$ (comp. Proposition A). Since $p_j \notin \partial U_i \ \forall j$, we have $\inf_{x \in U_i} d(x, p_j) > 0 \ \forall j \neq i$ and so the reciprocal $x \mapsto d(x, p_j)^{-1}$ is Lipschitz and bounded on $U_i$ (comp. Proposition B) as is the product with $x \mapsto \|y(x) - y(p_j)\|$ (comp. Proposition C) for all $j \neq i$. Define the following finite family of functions on $U_i$:

$$\mathcal{F}^i := \left\{ f_j^i(x) := \frac{\|y(x) - y(p_j)\|}{d(x, p_j)} \mid j = 1, \ldots, m \right\}$$

Combining the aforementioned facts, the functions $f_j^i$ are Lipschitz on $U_i$ (note that $f_i^i(x)$ vanishes on $U_i$). Let $L_j^i := L(f_j^i)$ be the corresponding Lipschitz constants and $K^i := \max_j L_j^i$. Define:

$$\varphi^i(x) := \sup\{f_j^i(x) \mid f_j^i \in \mathcal{F}_i\} = \max_j\{f_j^i(x)\}$$

As $f_j^i(x)$ is finite for every $x \in U_i$, so is $\varphi^i(x)$. Thus, $\varphi^i$ is Lipschitz on $U_i$ with Lipschitz constant $L(\varphi^i) = K^i$ (comp. Proposition D).

**Remark 1** Theorem 1 applies to $(M, \tilde{d})$ because it is a metric space and the continuity on $X$ w.r.t. $d$ implies the continuity of $y$ on $M$ w.r.t. $\tilde{d}$, see (Gluck, 1966).

We quickly verify that the information provided by $s_y$ is robust under label smoothing (Szegedy et al., 2015) with a uniform distribution. Let $\alpha \in (0, 1)$ and consider the smoothed labels $\hat{y} := (1 - \alpha)y + (\alpha C^{-1})1_C$, where $C$ is the number of classes and $1_C$ is the vector in $\mathbb{R}^C$ containing 1s in each component. Then the distance between labels is proportional to the previous for all $i, j$, more precisely, $\|\hat{y}(p_i) - \hat{y}(p_j)\| = (1 - \alpha)\|y(p_i) - y(p_j)\| \ \forall i, j$ which leads to the equivalent sensitivity information.

## APPENDIX B (ADDITIONAL EXPERIMENTAL DETAILS AND RESULTS)

### CALCULATING SENSITIVITY VALUES

Distributed over 32 processes, the calculations for a dataset of 50,000 elements with 3072 continuous features (specifications of the CIFAR-10 training set) with the $l_2$ norm take approximately 1 hour on the 32 cores of an AMD EPYC 7452.

---

**Algorithm 1** Calculating Sensitivity Values

---

**Require:** Dataset $\mathcal{D} := \{p_1, \ldots, p_m\}$ with labels $y_i := y(p_i)$ for $p_i \in \mathcal{D}$, metric $d$, empty list $L = []$

**for** $i = 1, \ldots, m$ **do**
   $M = -1$
   **for** $j = 1, \ldots, m$ **do**
     **if** $y_i \neq y_j$ **then**
       $M = \max\{M, \frac{1}{d(p_i, p_j)}\}$
     **else**
       *pass*
     **end if**
   **end for**
   Append $M$ to $L$
**end for**
**return** $L$

---

### HYPERPARAMETER GRID

The following hyperparameter grid with 729 possible settings was searched (after a brief manual exploratory phase to pinpoint sensible ranges); we used 20 diffusion steps. We refer to the paper of Karras et al. (2022) for a precise definition of these parameters.

- "sigma_min" : [0.003, 0.004, 0.005]
- "sigma_max" : [1.3, 1.4, 1.5]
- "rho" : [1.5, 1.6, 1.7]
- "S_churn" : [2.4, 2.5, 2.6]
- "S_min" : [0.4, 0.5, 0.6]
- "S_max" : [float('inf')]
- "S_noise" : [1.006, 1.012, 1.015]

### ADVERSARIAL ROBUSTNESS TRAINING DETAILS

Our evaluation setup for the adversarial robustness experiments matches the one used by Wang et al. (2023). We reiterate the details: the model architecture was a Wide-Res-Net (WRN) (Zagoruyko & Komodakis, 2016) with Swish & Silu activation functions (Hendrycks & Gimpel, 2016); due to resource constraints, we concentrated on the WRN-28-10 version with a batch size (BS) of 2048 (trained over 4 A100 NVIDIA 40GB GPUs). Adversarial training (Madry et al., 2018) was performed with the TRADES loss function (Zhang et al., 2019) either for the $l_2$ or $l_\infty$ norm (N), combined with label smoothing (Szegedy et al., 2015) (factor of 0.1) and weight averaging (Izmailov et al., 2018) with a decay rate of 0.995. We traced the adversarial accuracy on a separate set of 1024 elements to determine the "best" weights. The attacks for the training and tracking process used projected gradient descent (PGD) (Madry et al., 2018) with 10 and 40 steps, respectively, and maximum $l_2$ and $l_\infty$ perturbations of 128/255 and 8/255 with step sizes of 32/255 and 2/255, respectively. During training, we used stochastic gradient descent with Nesterov momentum (Nesterov, 1983) (momentum factor and weight decay were set to 0.9 and $5 \cdot 10^{-4}$, respectively) over a 1-cycle learning rate with warm-up and cosine annealing (Smith & Topin, 2019), where the

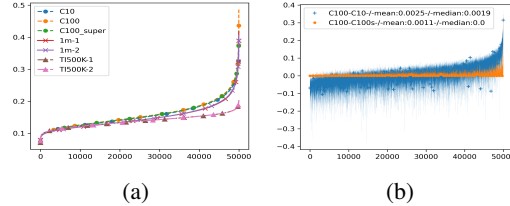

(a)  (b)

Figure 5: (a): Ordered $l_\infty$ sensitivity values for the original sets and the "TI500K" and "1m" data. (b): Differences (w.r.t. the $l_\infty$ norm) between sensitivity values of CIFAR-100/10 ("+") and CIFAR-100/100s ("o"), ordered by the CIFAR-100 values.

(maximum) learning rate was set to 0.2. We also used the additional CIFAR-10/100 datasets from Wang et al. (2023) with 20 (20m) and 50 (50m) million generated images (GI), which were mixed into each training batch at a rate of 80% (i.e. each batch consisted of 20% original images). The adversarial robustness was evaluated using the AutoAttack framework (Croce & Hein, 2020) for the "best" and the last weights after 2400 or 1600 epochs (E) of training, depending on the dataset/norm combination.

ADDITIONAL RESULTS FOR CIFAR

Here we display similar plots as in Fig. 2 and 3. We also added plots based on totally random trees (Geurts et al., 2006; Moosmann et al., 2006), also called random trees embeddings (*RTE*). In contrast to *MDS*, these fit a forest of random trees to the data points, whose representation is determined by the leaves they end up in (comp. `https://scikit-learn.org/stable/modules/generated/sklearn.ensemble.RandomTreesEmbedding.html`). Naturally, nearby points are more likely to fall into the same leaf.

Table 5: Pixel distances and FID between the 10,000 least/most robust images and their clones (O:C) and the pixel distances between 10 clones of the 1024 least/most robust images (C:C), respectively.

| Set | $\| \cdot \|$-Rob. | $l_\infty$ (O:C) | $l_2$ (O:C) | FID (O:C) | $l_\infty$ (C:C) | $l_2$ (C:C) |
|---|---|---|---|---|---|---|
| CIFAR-100 | $l_2$-Most | 0.64 | 7.76 | 6.08 | 0.69 | 8.61 |
| CIFAR-100 | $l_2$-Least | 0.49 | 6.17 | 6.14 | 0.50 | 6.72 |
| CIFAR-100 | $l_\infty$-Most | 0.64 | 7.65 | 5.99 | 0.65 | 7.94 |
| CIFAR-100 | $l_\infty$-Least | 0.48 | 6.29 | 6.29 | 0.49 | 6.81 |

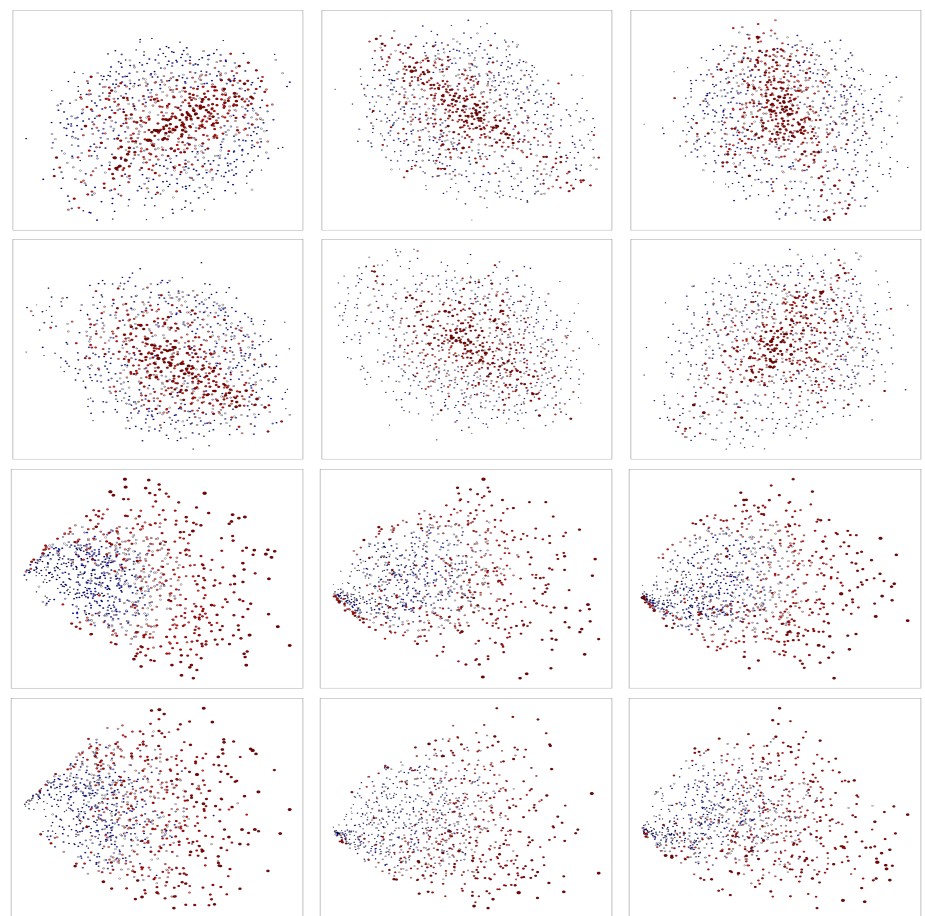

Figure 6: Top to bottom: *MDS* (rows 1,2) and RTE (rows 3,4) representations of the CIFAR-10/100/100s sets (left to right), coloured and resized in proportion to their $l_2$ (rows 1,3) and $l_\infty$ (rows 2,4) robustness values, where the most and least robust elements are the small blue and large red points, respectively.

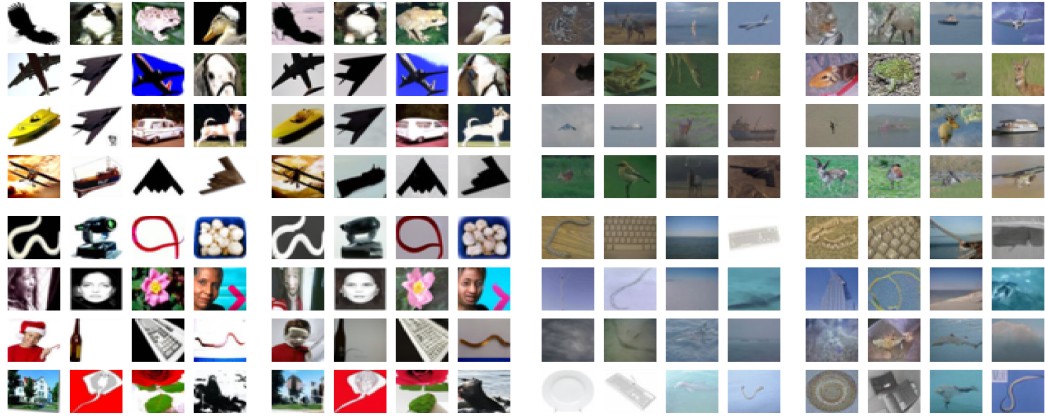

Figure 7: Top half: 16=4x4 most (square 1) and least robust elements (square 3) w.r.t. the $l_\infty$ norm from the CIFAR-10 set and their respective clones (squares 2,4). Bottom half: the corresponding plots for CIFAR-100.

## APPENDIX C (ADDITIONAL RESULTS FOR EMNIST)

The EMNIST set (Cohen et al., 2017) combines $28 \times 28$ greyscale images of digits and letters. It allows users to extract subsets containing either digits or lower- and upper-case letters with 10 +

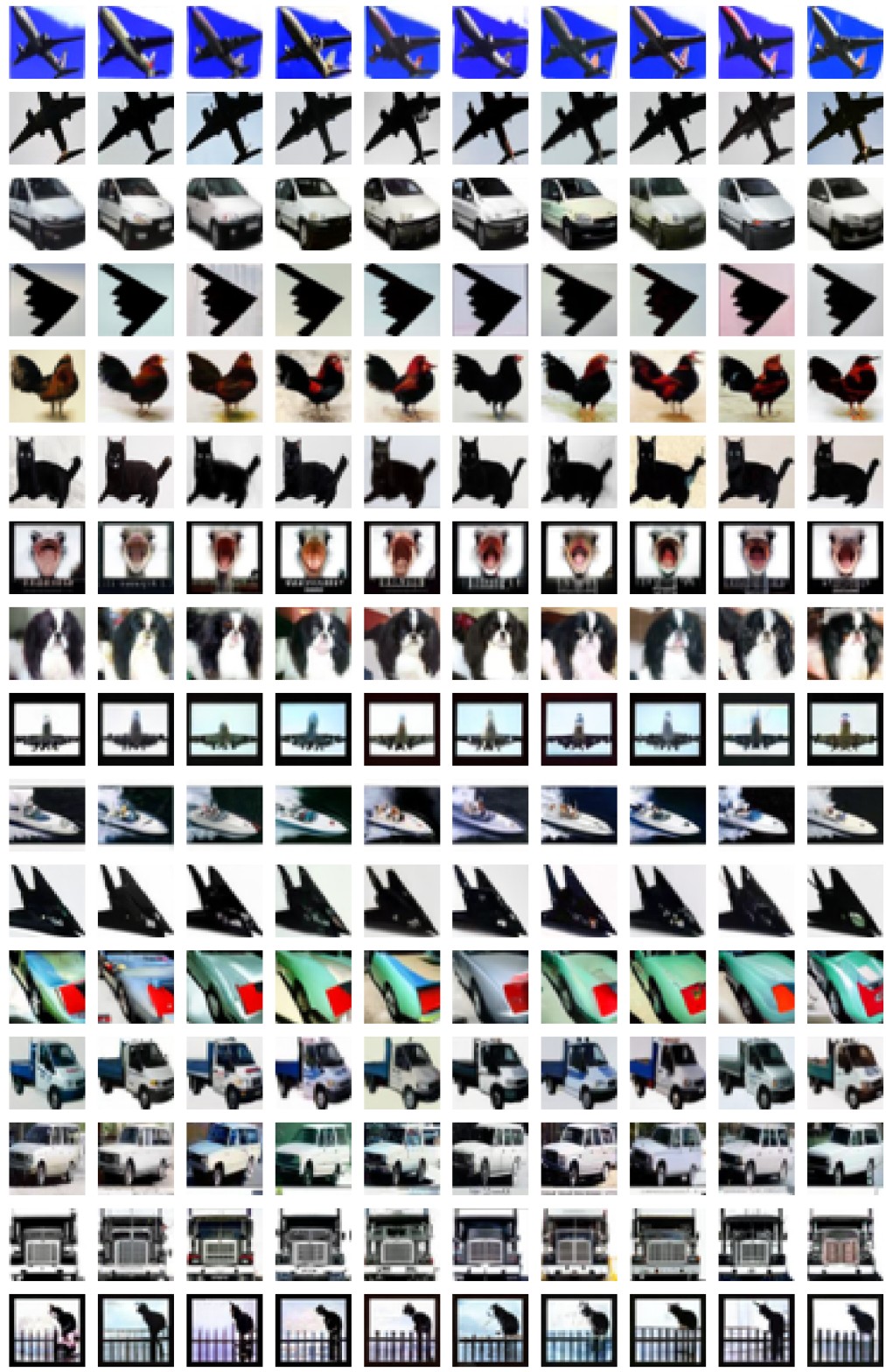

Figure 8: Ten cloned images for each of the 16 most robust elements from CIFAR-10 for the $l_2$ norm.

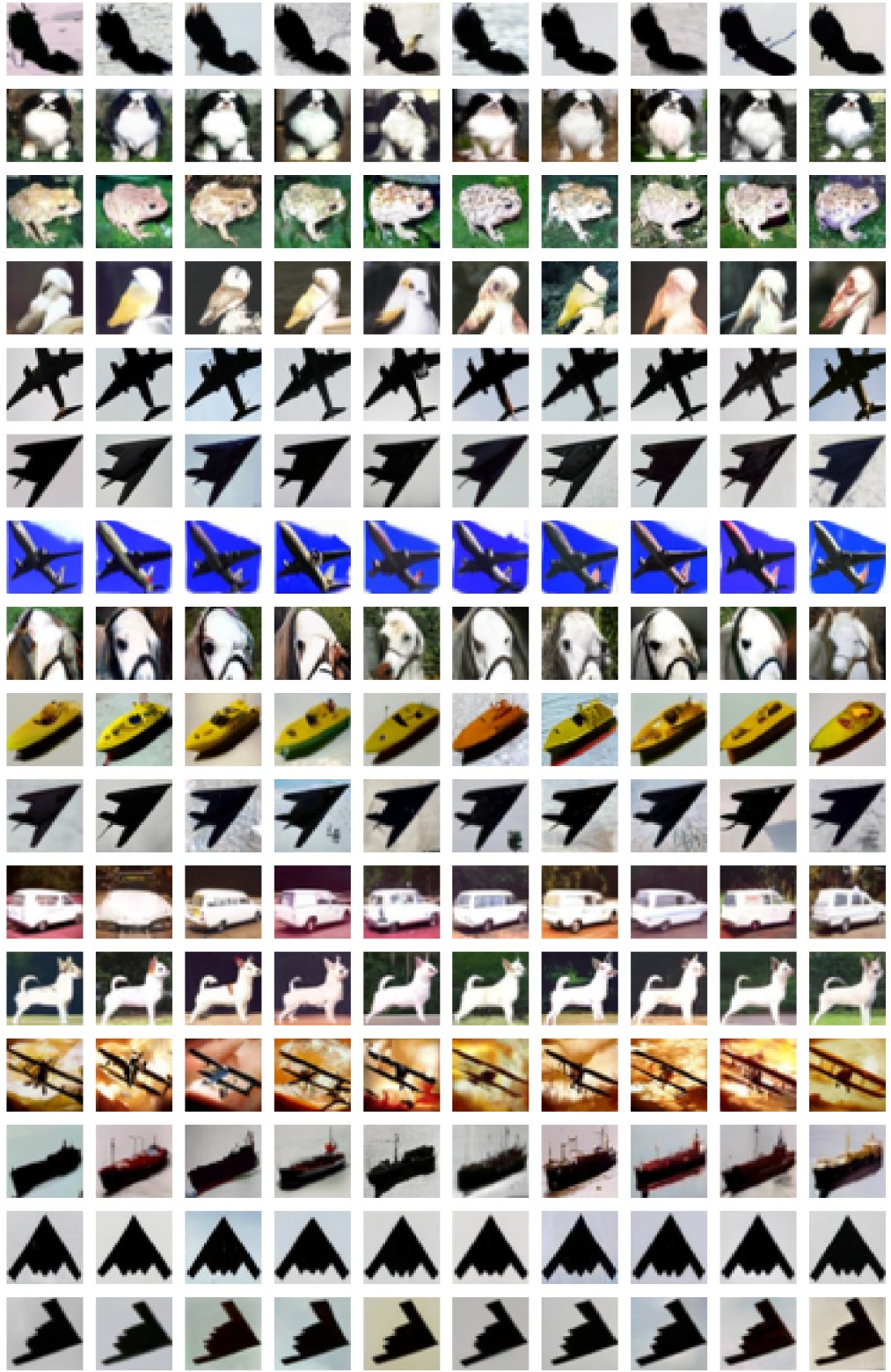

Figure 9: Ten cloned images for each of the 16 most robust elements from CIFAR-10 for the $l_\infty$ norm.

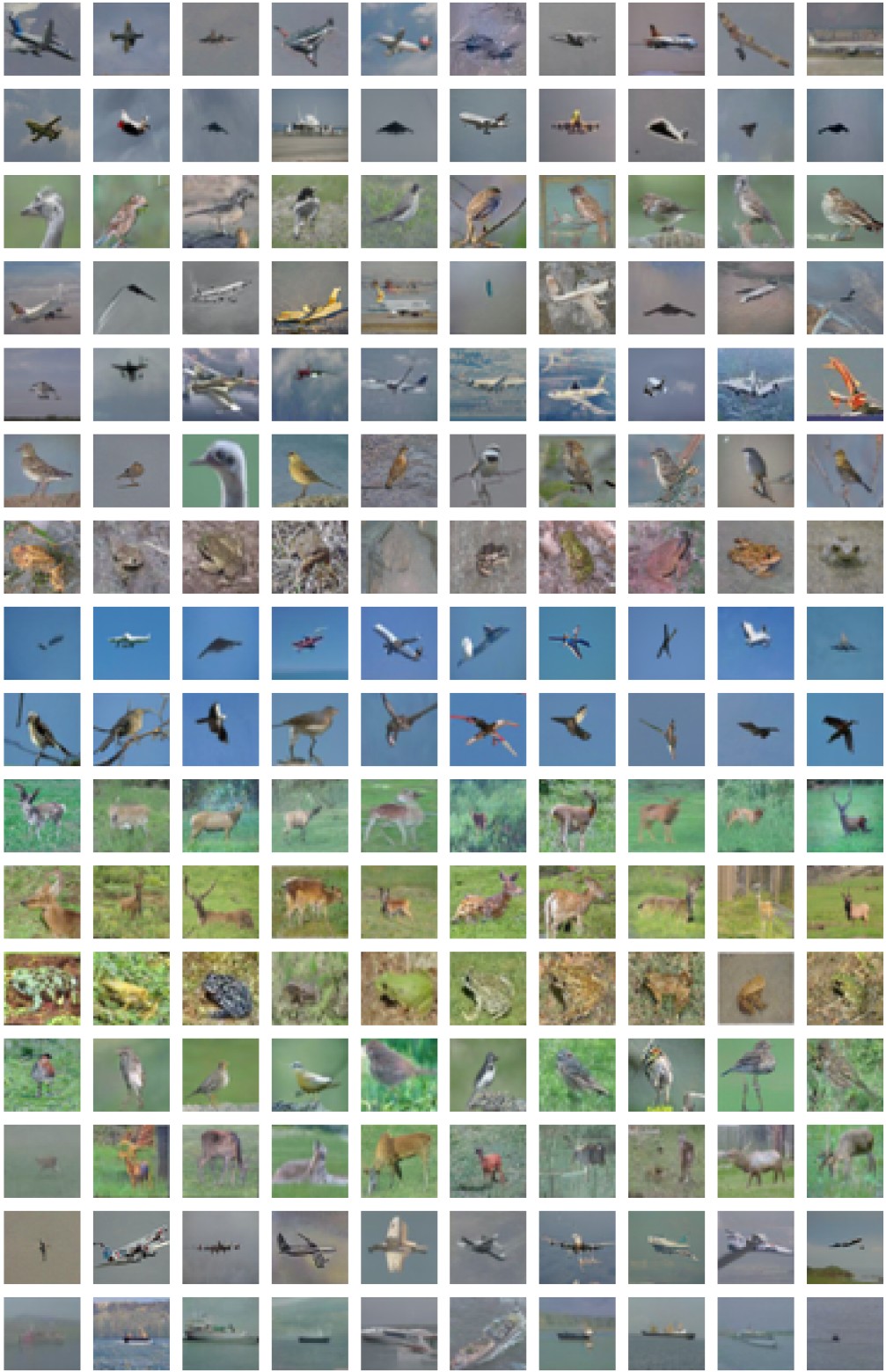

Figure 10: Ten cloned images for each of the 16 least robust elements from CIFAR-10 for the $l_2$ norm.

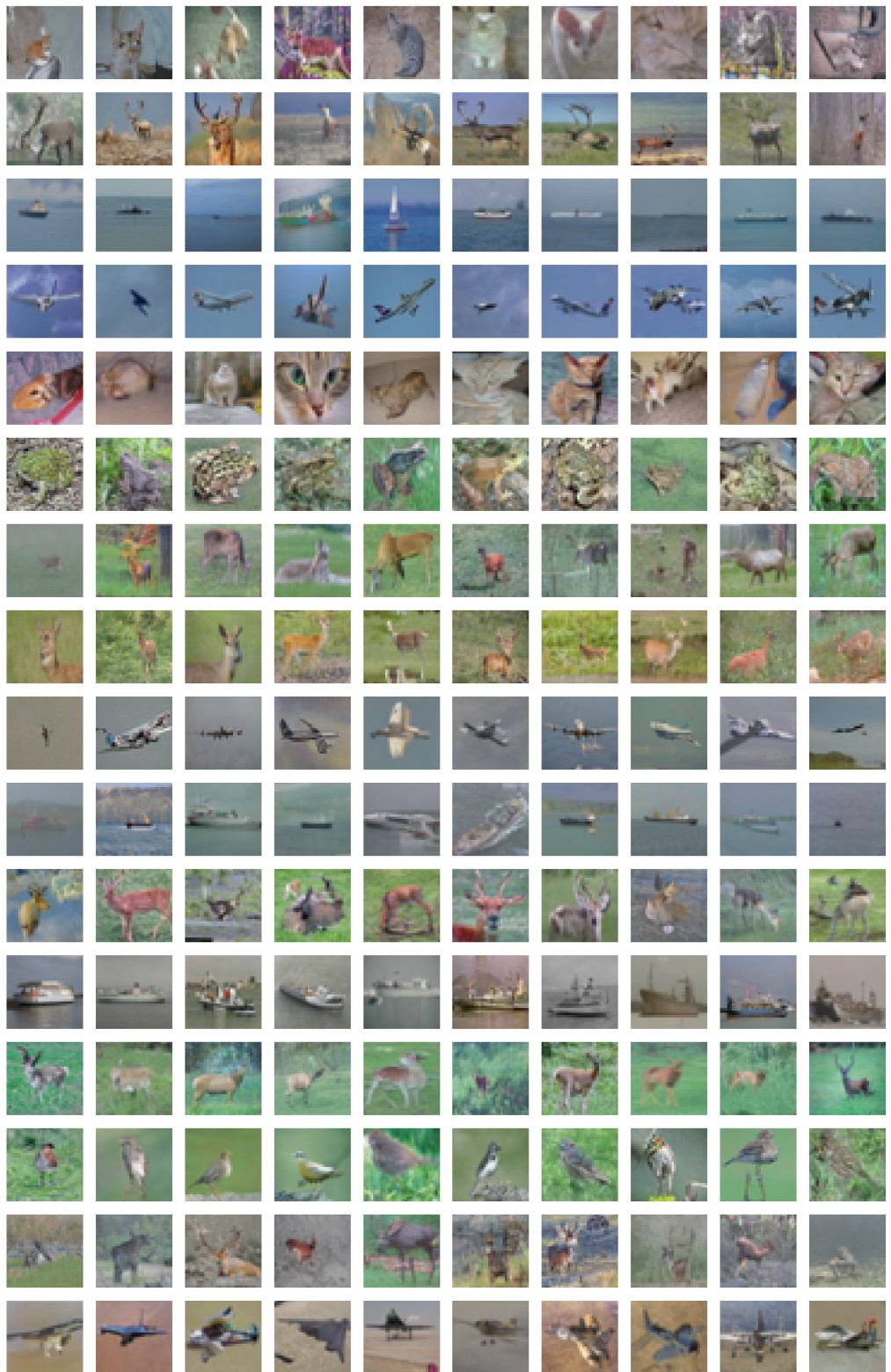

Figure 11: Ten cloned images for each of the 16 least robust elements from CIFAR-10 for the $l_\infty$ norm.

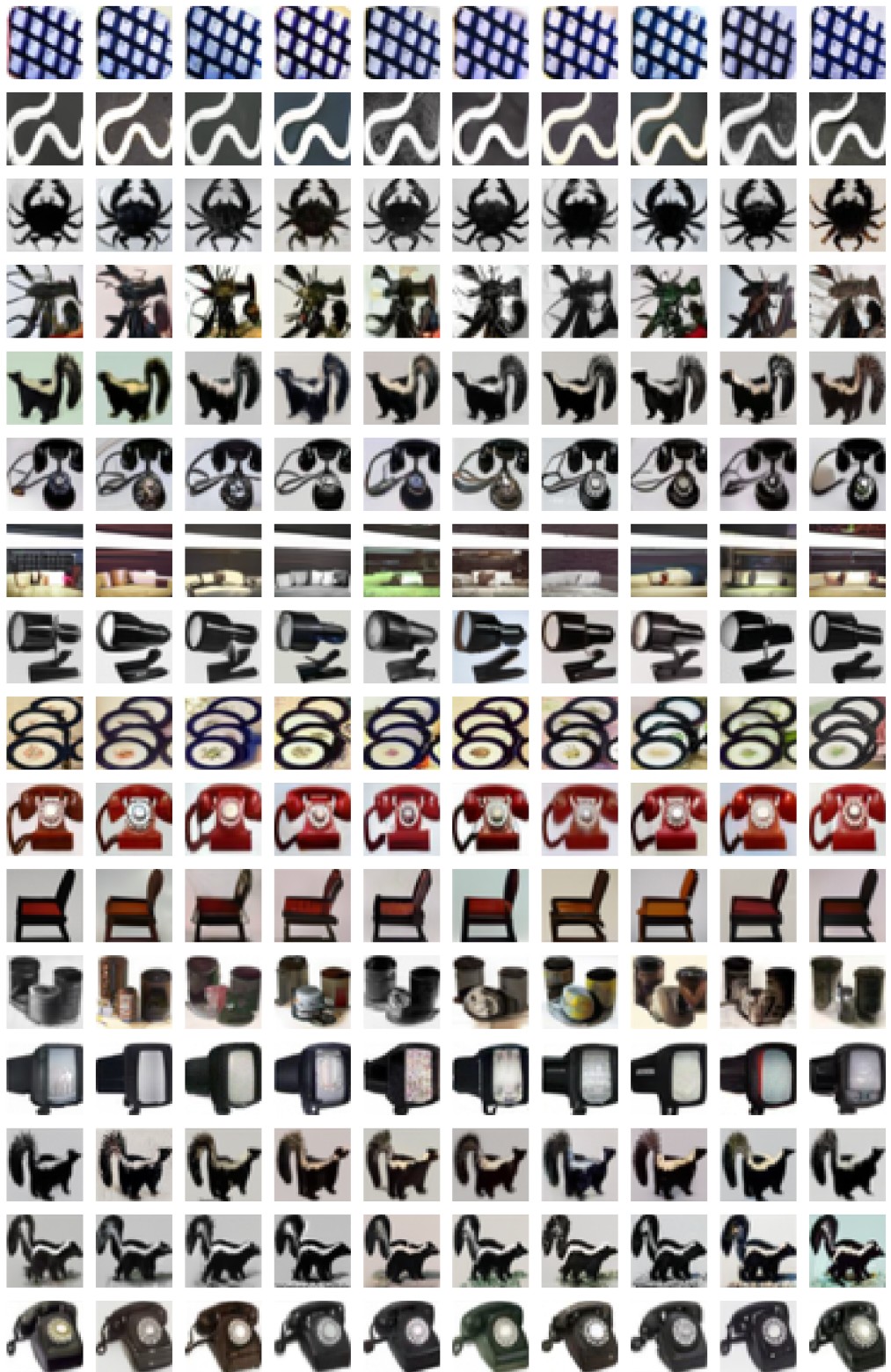

Figure 12: Ten cloned images for each of the 16 most robust elements from CIFAR-100 for the $l_2$ norm.

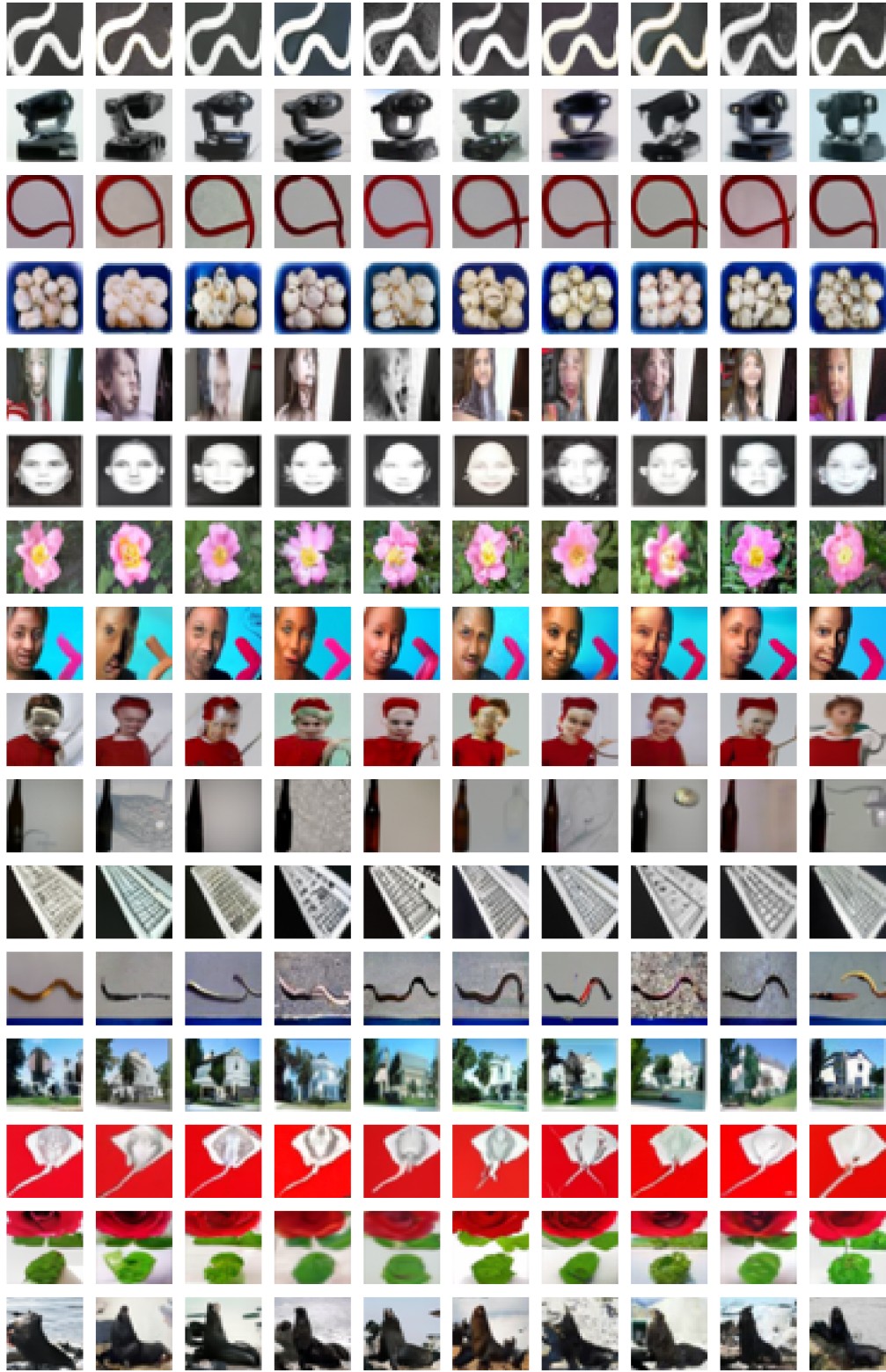

Figure 13: Ten cloned images for each of the 16 most robust elements from CIFAR-100 for the $l_\infty$ norm.

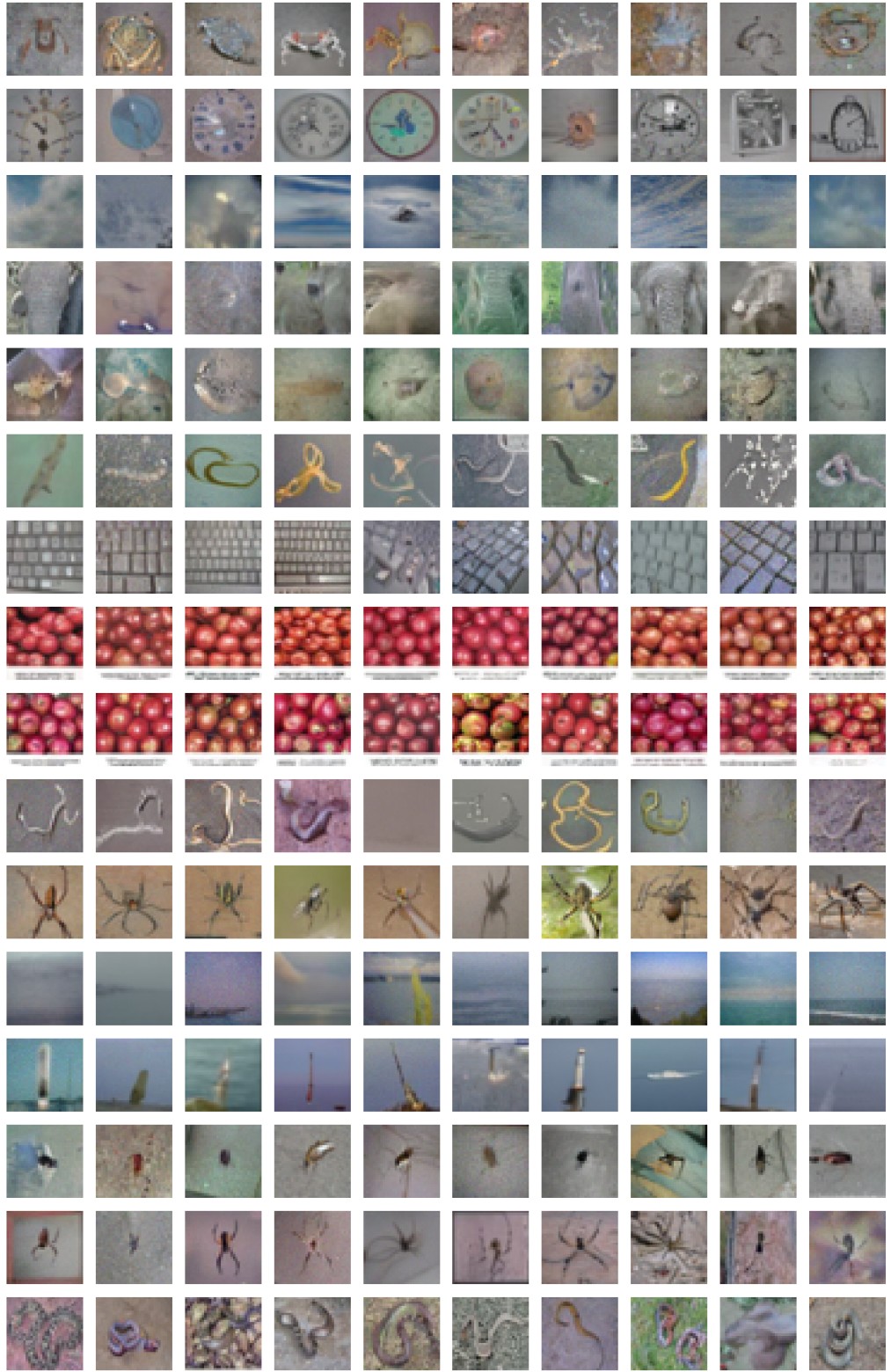

Figure 14: Ten cloned images for each of the 16 least robust elements from CIFAR-100 for the $l_2$ norm.

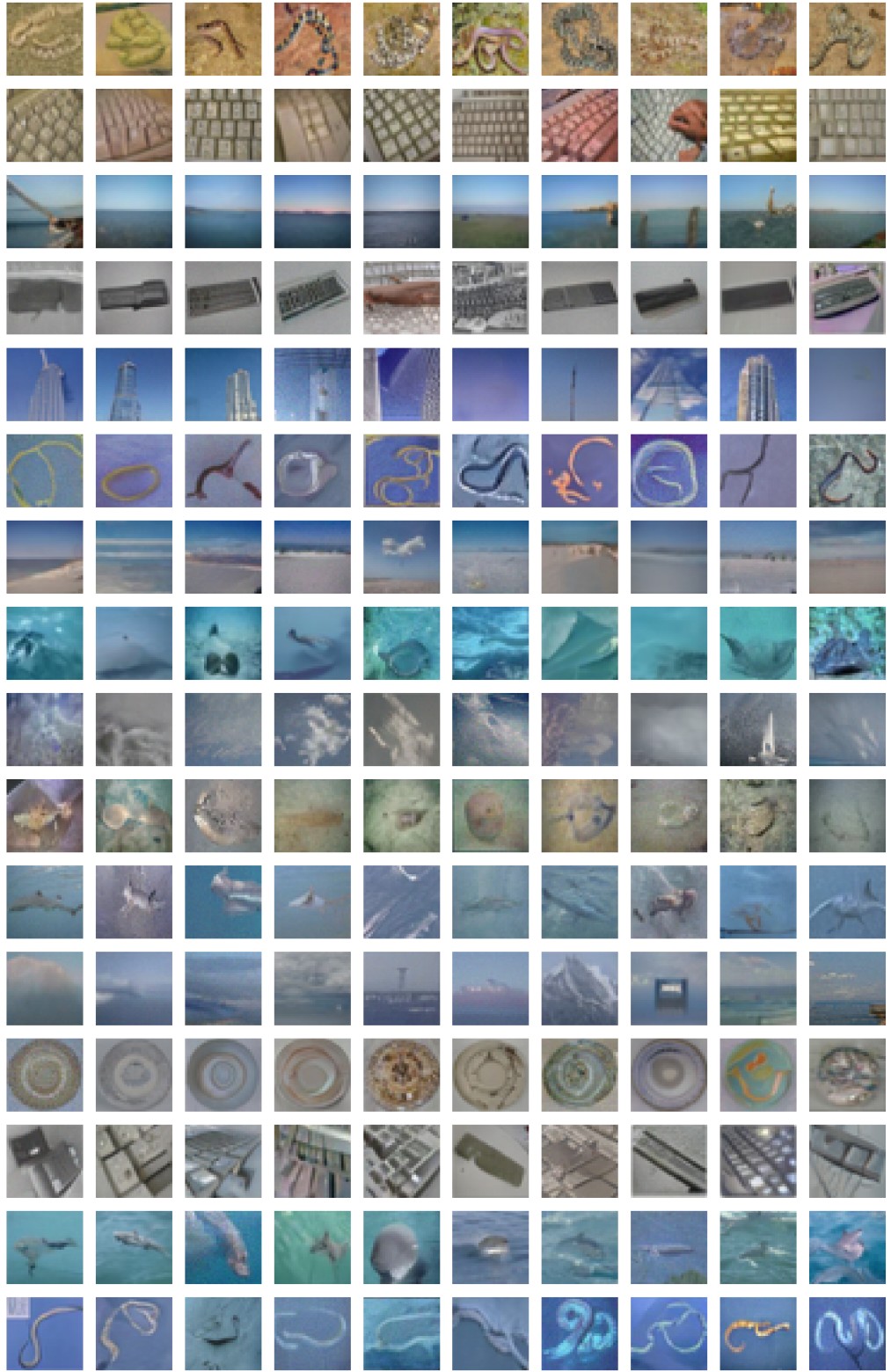

Figure 15: Ten cloned images for each of the 16 least robust elements from CIFAR-100 for the $l_\infty$ norm.

26 + 26 = 62 classes. We use the following definitions (again concentrating on the training sets): $M$:=MNIST (a subset of only digits; 10 classes; 60,000 elements), $D$:=Digits (only digits, but more elements; 10 classes; 240,000 elements), $L$:=Letters (merged upper-/lowercase letters; 26 classes; 88,800 elements), $B$:=Balanced (balanced mix of digits and upper-/lowercase letters, where some lowercase letters are regarded as uppercase; 47 classes; 112,800 elements), $BM$:=By-Merge (mix of all digits and upper-/lowercase letters where some lowercase letters are regarded as uppercase; 47 classes; 697,932 elements), $BC$:=By-Class (mix of all digits and upper-/lowercase letters; 62 classes; 697,932 elements). We refer to the paper of Cohen et al. (2017) for more details.

Figure 16 displays the ordered absolute (left half) and relative (right half) $l_2$ (plots 1,3) and $l_\infty$ (plots 2,4) sensitivity distributions of the six EMNIST (sub)sets. Since the sets have different numbers of elements, we normalised the plots on the horizontal axis; as for CIFAR-100, we removed a small amount (40) of the least robust elements to avoid distortion. The distributions for the sets containing only digits ($M$, $D$) are similar, as are those containing both all digits and all lower- and uppercase letters ($BM$, $BC$). Interestingly, the distributions for the sets containing only letters ($L$) and a balanced mix of letters and digits ($B$) are also very similar w.r.t. each other but differ from the former distributions. As for the CIFAR data, the most significant changes are seen for the least robust elements (towards the right in plots 1 and 2).

Figure 17 displays the *MDS* and *RTE* representations w.r.t. the $l_2$ and $l_\infty$ norm, respectively, for samples of the three EMNIST subsets containing only digits ($M$), only letters ($L$) and a balanced mix of both ($B$). We again notice patterns emerging, although they are different in shape compared to the CIFAR representations. In particular, clusters of the least robust (large red) points dominate the graphs, specifically for the $l_\infty$ norm.

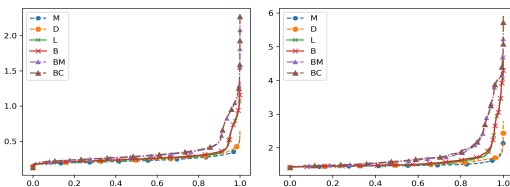

Figure 16: Ordered absolute $l_2$ (left) and $l_\infty$ sensitivity values of the six EMNIST (sub)sets.

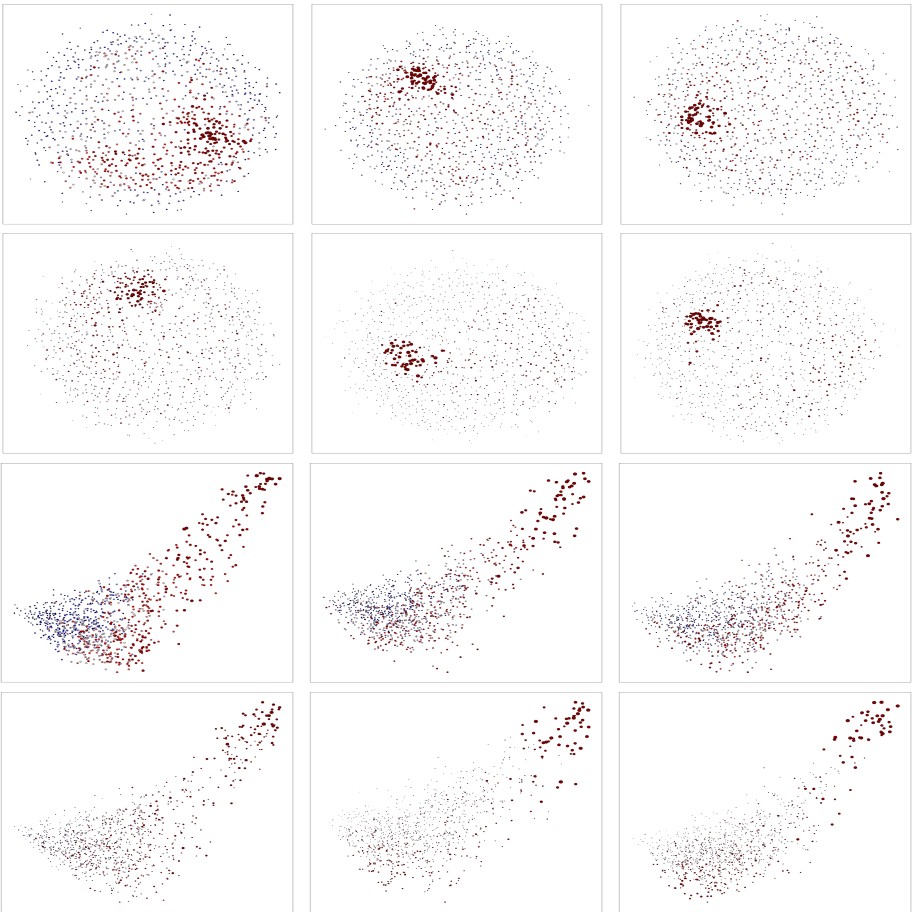

Figure 17: Top to bottom: *MDS* (rows 1,2) and RTE (rows 3,4) representations of the three EMNIST subsets *M*, *L*, *B* (left to right), coloured and resized in proportion to their $l_2$ (rows 1,3) and $l_\infty$ (rows 2,4) robustness values, where the most and least robust elements are the small blue and large red points, respectively.

## APPENDIX D (THE GLITCH)

We likely encountered a glitch in the `pytorch`-based (https://pytorch.org/) training pipeline, where information from the stronger PGD-40 attack seemed to leak into the training process whenever the small hold-out set was also included in the training data. This is peculiar because both were intended to be separate processes (using a context manager, see below). This allowed us to include the information from the stronger attacks on the focused set in the optimisation process of the following training epoch. The validation call and the code snippet intended to prevent this (as was the original idea) are shown in Figures 18 and 19 from the `gowal21uncovering/utils/watrain.py` and `core/utils/context.py` scripts, respectively, taken from the codebase at https://github.com/wzekai99/DM-Improves-AT/tree/main.

If the `adversarial` argument is true in the `eval` function (Figure 18, line 1), the model will be attacked on the samples contained in the `dataloader` and its adversarial accuracy is returned. In this case, the `ctx_noparamgrad_and_eval` context manager is called (Figure 18, line 11) which combines the functions `ctx_noparamgrad` (Figure 19, line 1) and `ctx_eval` (Figure 19, line 12). The purpose of these two functions is to

- save the original gradient (Figure 19, line 3) and training states (Figure 19, line 14) of the parameters and the modules of the model,

- initialise class-dependent instances of the modules (Figure 19, line 4, 15),

- set the `requires_grad` (Figure 19, lines 5, 48) and the `training` (Figure 19, line 16, 37) argument of the parameters and modules to `False`,

- and return the model in its previous state upon exiting after the attack was performed (Figure 19, lines 8, 19).

However, whereas `ctx_noparamgrad_and_eval` yields `self.module` (Figure 19, line 26), the deactivation calls are applied to `module` which does not affect `self.module` (we tested that these are separate entities). We *think* that this has the effect of the eval-process accumulating gradient information of the parameters which are not explicitly deleted afterwards but rather picked up by the automatic differentiation mechanism. Since the latter builds computation graphs from the inputs to the outputs in the forward pass (see `https://pytorch.org/blog/computational-graphs-constructed-in-pytorch/`), the gradient information may be reused in the following training epoch in the backward pass **if** the elements appear again in any of the training batches. However, as mentioned before, we were able to circumvent this assumed glitch by coding the outlined mechanisms explicitly. More precisely, we excluded the context manager, crafted the stronger PGD-40 adversarial examples on the focused sets in the beginning of each epoch and included them in each training batch at a rate of 32 (the delta when increasing the batch size from 2048 to 2080).

```python
def eval(self, dataloader, adversarial=False):
    """
    Evaluate performance of the model.
    """
    acc = 0.0
    self.wa_model.eval()

    for x, y in dataloader:
        x, y = x.to(device), y.to(device)
        if adversarial:
            with ctx_noparamgrad_and_eval(self.wa_model):
                x_adv, _ = self.eval_attack.perturb(x, y)
            out = self.wa_model(x_adv)
        else:
            out = self.wa_model(x)
        acc += accuracy(y, out)
    acc /= len(dataloader)
    return acc
```

Figure 18: Evaluation function call.

```python
class ctx_noparamgrad(object):
    def __init__(self, module):
        self.prev_grad_state = get_param_grad_state(module)
        self.module = module
        set_param_grad_off(module)    #<-The argument should be <self.module>.
    def __enter__(self):
        pass
    def __exit__(self, *args):
        set_param_grad_state(self.module, self.prev_grad_state)
        return False

class ctx_eval(object):
    def __init__(self, module):
        self.prev_training_state = get_module_training_state(module)
        self.module = module
        set_module_training_off(module)    #<-The argument should be <self.module>.
    def __enter__(self):
        pass
    def __exit__(self, *args):
        set_module_training_state(self.module, self.prev_training_state)
        return False

@contextmanager
def ctx_noparamgrad_and_eval(module):
    with ctx_noparamgrad(module) as a, ctx_eval(module) as b:
        yield (a, b)

def get_module_training_state(module):
    return {mod: mod.training for mod in module.modules()}

def set_module_training_state(module, training_state):
    for mod in module.modules():
        mod.training = training_state[mod]

def set_module_training_off(module):
    for mod in module.modules():
        mod.training = False

def get_param_grad_state(module):
    return {param: param.requires_grad for param in module.parameters()}

def set_param_grad_state(module, grad_state):
    for param in module.parameters():
        param.requires_grad = grad_state[param]

def set_param_grad_off(module):
    for param in module.parameters():
        param.requires_grad = False
```

Figure 19: Context manager function calls.

