# OpenReview forum: "Advancing the Adversarial Robustness of Neural Networks from the Data Perspective"
_ICLR.cc/2024/Conference — Submitted to ICLR 2024_

### Official Review · Reviewer_pfh7 · 2023-10-29

**Soundness:** 2 fair
**Presentation:** 3 good
**Contribution:** 2 fair
**Rating:** 3
**Confidence:** 3

**Summary:**

The authors establish a connection between the curvature of the data manifold, as perceived by a model during training, and the model’s adversarial robustness. They provide empirical evidence showing that neural networks gain adversarial robustness more slowly in less robust regions of the data manifold.

**Strengths:**

- A novel perspective on adversarial robustness through the lens of data robustness (curvature of data manifold)
- Claims are backed up by empirical results
- Generally good writing

**Weaknesses:**

The most critical issue is the lack of comparison with prior works. It is well-known in the literature that not all data points are equally susceptible to adversarial attack, and this has motivated the design of various variants of adversarial training ([1,2,3,4,5,6]) to (adaptively) focus on a subset of training samples. The authors made a similar observation "it appears beneficial to emphasize the least robust elements during training", but seemed to be completely unaware of this line of research. Without proper discussion and comparison with prior works, it is hard to fairly position and evaluate this work in the vast literature.

References

[1] Cai, Qi-Zhi, Chang Liu, and Dawn Song. "Curriculum adversarial training." Proceedings of the 27th International Joint Conference on Artificial Intelligence. 2018.

[2] Ding, Gavin Weiguang, et al. "MMA Training: Direct Input Space Margin Maximization through Adversarial Training." International Conference on Learning Representations. 2019.

[3] Wang, Yisen, et al. "On the Convergence and Robustness of Adversarial Training." International Conference on Machine Learning. PMLR, 2019.

[4] Zhang, Jingfeng, et al. "Geometry-aware Instance-reweighted Adversarial Training." International Conference on Learning Representations. 2020.

[5] Zeng, Huimin, et al. "Are adversarial examples created equal? A learnable weighted minimax risk for robustness under non-uniform attacks." Proceedings of the AAAI Conference on Artificial Intelligence. Vol. 35. No. 12. 2021.

[6] Xu, Yuancheng, et al. "Exploring and Exploiting Decision Boundary Dynamics for Adversarial Robustness." The Eleventh International Conference on Learning Representations. 2022.

[7] Xiao, Jiancong, et al. "Understanding adversarial robustness against on-manifold adversarial examples." arXiv preprint arXiv:2210.00430 (2022).

**Questions:**

One of the mainstream hypotheses regarding adversarial examples is the off-manifold assumption ([7]): "Clean data lies in a low-dimensional manifold. Even though the adversarial examples are close to the clean data, they lie off the underlying data manifold." I would like to understand how this hypothesis is related to your findings (focusing on data manifold with high curvature is helpful in adversarial training).

---

> ### Author Response · Authors · 2023-11-21
> **Answer to Official Review of Submission3684 by Reviewer pfh7**
>
> ## *We thank reviewer "pfh7" for the criticism and the recommended literature.*
>
> ---
> ---
> ---
>
> ## ---*First, we address the mentioned weaknesses regarding the lack of comparison with prior works to delineate our contribution.*---
> ---
> There is indeed a vast corpus of literature regarding adversarial machine learning in general and adversarial training in particular. In order to (re)position our work with respect to the listed papers, we have revised our exposition to include the previous advancements and clarify which “gap” we intend to close with our contribution. More precisely, we focus on “why“ adversarial training shows a weaker effect in regions of non-robust data which cannot be explained entirely by small margins between differently labelled points.
>
> To stress this point, we added to the updated version that the minimal $l_\infty$ distance between two non-robust CIFAR-10 images ($\approx0.2118$) is more than six times as large as the maximum perturbation radius ($\approx0.0314$) which, geometrically, leaves more than enough space for a robust decision boundary. However, it is not found consistently in these regions, as seen in Fig. 1.
>
> Our primary goal and contribution in this work is to provide a new perspective on this phenomenon which builds on a deeper geometric concept, namely curvature (as perceived by a model during training). Whereas one part of this contribution is the theoretical framework connecting the robustness of data to the perceived curvature, our empirical results show that this perspective is meaningful in practice.
> Regarding the adversarial training experiments, this view of perceived curvature can explain why focusing on a negligible number of non-robust points during training leads to an increased robust generalisation accuracy beyond the expected local improvements. Indeed, as we clarify in the updated version, the focus on the least robust points entails a better understanding of the pixel distances (the local improvement) and a better understanding of semantic distances (judging by the improved performance on unseen data). Together, this leads to a better understanding of curvature.
>
> Whereas we do improve upon the current state-of-the-art approach from Wang et al. (2023) in a cost-neutral way, the main contribution of our work is a deeper understanding of representation learning. Our focus is not to provide a new method of adversarial training (which is beyond the scope of this work) but to motivate why developing methods based on this deeper geometric insight is promising in the first place.
>
> ---
> ---
> ---
> ## ---*Regarding the question.*---
> ---
> First, if shifting points away from the true data manifold alone would impede a model's performance, we would expect a much worse accuracy on data augmented with, for example, slight uniformly distributed noise. (It is mathematically impossible to generate noise tangent to the manifold without knowing its shape.) However, as shown recently (Xiao et al. (2022); Chen et al. (2023)), it is possible to generate (unrestricted) semantic adversarial examples which are not only close but on the data manifold.
>
> What the view of a model perceiving curvature allows is to combine both types of vulnerabilities described above. Adversarial noise normal to the data manifold is detrimental even when the model's internal representation of the data concurs with the true representation. On the other hand, adversarial variations of the data may align with the data manifold if the model's learned representation of the latter is distorted in the first place. Our proposed view of a model perceiving curvature can explain how and where these distorted representations appear and why, for example, adversarial training shows a weaker effect in the least robust regions.
>
> Intuitively, in the regions of perceived high curvature, the training algorithm forces a model to connect a binary "semantic" change (labels are either different or not) to a relatively small change in terms of pixels (as is the case in the least robust regions). This may lead to the model adapting by learning spurious correlations to account for this perceived imbalance between semantic and pixel variation, ultimately making the model more susceptible to (adversarial) noise. This can explain why models can even be susceptible to noise perturbations tangent to the data manifold (because they are not aligned with the model's internal representation of the same).

---

> > ### Author Response · Authors · 2023-11-21
> > **Answer to Official Review of Submission3684 by Reviewer pfh7 (continued)**
> >
> > ---
> > ---
> > ---
> > ## ---*References*---
> > ---
> > Zekai Wang, Tianyu Pang, Chao Du, Min Lin, Weiwei Liu, and Shuicheng Yan. Better diffusion models
> > further improve adversarial training. In Andreas Krause, Emma Brunskill, Kyunghyun Cho, Barbara
> > Engelhardt, Sivan Sabato, and Jonathan Scarlett (eds.), Proceedings of the 40th International
> > Conference on Machine Learning, volume 202 of Proceedings of Machine Learning Research, pp.
> > 36246–36263. PMLR, 23–29 Jul 2023.
> >
> > ---
> >
> >  Xiao, Jiancong, et al. "Understanding adversarial robustness against on-manifold adversarial examples." arXiv preprint arXiv:2210.00430 (2022).
> >
> > ---
> >
> > Xinquan Chen, Xitong Gao, Juanjuan Zhao, Kejiang Ye, Cheng-Zhong Xu; Proceedings of the IEEE/CVF International Conference on Computer Vision (ICCV), 2023, pp. 4562-4572

---

### Official Review · Reviewer_vUSY · 2023-11-03

**Soundness:** 3 good
**Presentation:** 2 fair
**Contribution:** 3 good
**Rating:** 5
**Confidence:** 3

**Summary:**

The paper leverages concepts from metric geometry to understand how neural networks perceives the geometric property, particularly curvature, of the data manifold. To be more specific, it argues that data that more susceptible to adversarial perturbations are connected to regions in the data manifold with high curvature and vice versa. The paper proposes to use Eq 1 to quantitatively measure such curvature information as perceived by the model. The empirical studies in this paper are based on CIFAR10 and CIFAR100. A series experiments are performed to verify the proposed connection between curvature and model robustness. Building on these findings, the paper propose a learning rate strategy that increases the adversarial robustness model against $\ell_2$ and $\ell_\infty$-norm bounded adversarial perturbations generated using AutoAttack in a white-box setting.

**Strengths:**

**Originality**:
The paper proposes a very novel (to my knowledge) and unique perspective to understand the adversarial robustness of neural networks. The proposed concept is also quite intuitive: certain data points are inherently more susceptible to adversarial attacks due to specific properties they possess.

**Quality**:
I am not an expert in manifold learning nor in metric geometry, so this limit my ability to properly assess the technical details of Section 3.  However, on the empirical side, the paper tries to provide several experiments to validate their hypothesis regarding the connection between model's robustness and its perception on the curvature of data manifold.

**Significance**:
The proposed data perspective provides an interesting direction on which future methods can be designed to better address the adversarial robustness problem.

**Weaknesses:**

The paper has two main weaknesses.

1. The clarity of the presentation and the quality of interpretation regarding the empirical observations could benefit from further refinement to enhance understanding.
2. The improvement in adversarial robustness is very marginal.

**Questions:**

**Figure 1**:
I suppose this is on CIFAR10. How are the most/east robust elements defined?
The claim at the bottom of page 1 is essentially that by training for another 2000 epochs, the most robust training data (at epoch 400) become 1% more robust (at epoch 2400); and the least robust training data (at epoch 400) becomes 7.5% more robust (at epoch 2400). Is that correct? However, are we tracking the same training data? Is "eval_adversarial_acc" the validation accuracy?

Also, it seems that the author uses the term "validation set" as a parts of the training set, which is odd.

I did not understand "For CIFAR-100, we disregarded the 40 least robust elements because of distorted robustness values due to differently labelled duplicate images". What are "differently labelled duplicate images"? What happens to the result if we include them?


**Figure 2**:
My understanding is that the absolute sensitivity is computed using Eq1. How are the relative sensitivity computed?
In the analysis of Figure2, it is said that "The generated data, in particular, admit much shorter tails, indicating more robust elements overall. " However, arent the curves of  1m-1 and 1m-2 quite similar to cifar-10/100/100s? Also, why does the pseudo-labbeled data have very different sensitivity?

**Figure 3**:
The interpretation of the results are not clear. Do we know what the differences in the mechanisms of MDS and RTE are that leads to this inverted pattern? It would be very helpful to interpret the result if there is some brief explanation on what those methods are and why they are used.

**General suggestions on all figures**:
Please consider using subcaptions to increase the clarity of the results.

**Exploration experiments**:
In the setup column of Table2, just to clarify: minus v, "-v" means removing v from the training set, and (v) means all the validation accuracy is based on v. Is this correct?
Are the most sensitive data points computed based on another pre-trained model?

---

> ### Author Response · Authors · 2023-11-21
> **Answer to Official Review of Submission3684 by Reviewer vUSY**
>
> ## *We thank reviewer "vUSY" for the criticism and the suggestions to improve our work.*
> ---
> ---
> ---
> ## ---*First, we address the mentioned weaknesses individually.*---
> ---
> > *The clarity of the presentation and the quality of interpretation regarding the empirical observations could benefit from further refinement to enhance understanding.*
>
> To clarify the presentation of empirical results, we rephrased and refined Section 4 to make it less convoluted and more accessible. We focused on more intuitive descriptions, especially for readers who are not necessarily familiar with the topics.
>
> > *The improvement in adversarial robustness is very marginal.*
>
> In absolute terms, an improvement of 0.2 percentage points seems small but, as stated in the paper, amounts to the same increase of extending the training by 800 epochs, that is, one-third of the entire training (based on results from Wang et al. (2023)). Importantly, this benefit is achieved using identical computational resources.
> However, the primary goal of these experiments is to show that focusing on a negligible number of points from non-robust regions leads to an increased robust generalisation accuracy beyond the expected local improvements. In other words, a better understanding of the pixel distances between points is paired with a better understanding of semantic distances (hence the improved performance on unseen data).
>
>
> ---
> ---
> ---
> ## *Regarding the questions.*
> ---
>
>
>
> > *Figure 1: I suppose this is on CIFAR10. How are the most/east robust elements defined?*
>
>
> The robustness of a data point is the minimal distance to a differently labelled data point. We added an explicit definition in Section 3 to avoid confusion. However, the absolute robustness (or, reciprocally, sensitivity) value is of secondary importance. It is the order of points when sorted according to their robustness (or sensitivity) values which lets us connect these concepts to the perceived curvature of a model. For this reason, we always refer to the "least and most robust points" instead of "points with a robustness of <x> or smaller/larger than <x>". To make the concept more accessible to readers, we changed the introduction by taking a more intuitive approach to explaining data robustness.
>
>
>
>
> > *The claim at the bottom of page 1 is essentially that by training for another 2000 epochs, the most robust training data (at epoch 400) become 1% more robust (at epoch 2400); and the least robust training data (at epoch 400) becomes 7.5% more robust (at epoch 2400). Is that correct?*
>
> Correct.
>
>
> > *However, are we tracking the same training data? Is "eval_adversarial_acc" the validation accuracy?*
>
>
> The "eval_adversarial_acc" traces the adversarial robustness against a PGD-40 attack with maximum perturbation radius 8/255 and step size 2/255 on the 1024 least and most robust elements, respectively. We added more information to the introduction to clarify this.
> However, these subsets are also fixed parts of the training data in every epoch, allowing us to enforce a focus on these points during adversarial training. This example demonstrates that the effect of adversarial training is much weaker in non-robust regions compared to robust regions.
> We modified the exposition of this example by (i) removing two plots and (ii) changing its explanation, where we chose a more natural way to introduce data robustness beforehand.
>
>
>
>
>
>
>
>
>
>
>
> > *Also, it seems that the author uses the term "validation set" as a parts of the training set, which is odd.*
>
> To avoid confusion, we changed the nomenclature at several points.
>
>
>
>
>
>
>
> > *I did not understand "For CIFAR-100, we disregarded the 40 least robust elements because of distorted robustness values due to differently labelled duplicate images". What are "differently labelled duplicate images"? What happens to the result if we include them?*
>
> As we found out, the CIFAR-100 set contains identical images with different labels. For such data points, the sensitivity is not defined due to the vanishing quotient (comp. Equation 1). Based on the implementation, the sensitivity computation for these points would either return “Nan”s or extremely large values (as for us), both of which are inadmissible and distort the sensitivity distributions.
>
>
>
>
>
>
> > *Figure 2: My understanding is that the absolute sensitivity is computed using Eq1. How are the relative sensitivity computed?*
>
> The relative sensitivity distributions are empirical probability distributions over the absolute sensitivity values (similar to histograms). However, since these plots confuse readers (and add barely any meaningful information), we removed them.

---

> > ### Author Response · Authors · 2023-11-21
> > **Answer to Official Review of Submission3684 by Reviewer vUSY (continued)**
> >
> > > *In the analysis of Figure2, it is said that "The generated data, in particular, admit much shorter tails, indicating more robust elements overall. " However, arent the curves of 1m-1 and 1m-2 quite similar to cifar-10/100/100s? Also, why does the pseudo-labbeled data have very different sensitivity?*
> >
> >
> > This was indeed a mistake in the description of the graph, which we corrected. We also added an explanation as to why pseudo-labeled data show (comparatively) different sensitivity values. From Carmon et al. (2019), we know that these data points are “mined” from the 80 Million Tiny Images (80M-TI) dataset (Torralba et al., 2008), which is the same source as for the CIFAR images (Krizhevsky, 2009). Similar to the generated data from Wang et al. (2023), the labels were assigned using the classifier predictions with the highest confidence. However, Carmon et al. (2019) removed near-duplicates of the CIFAR-10 test set from the 80M-TI set in a preliminary phase which can explain the lack of highly sensitive elements. More precisely, as sensitivity is a local concept (proven in Theorem 1), removing near-duplicates can decrease sensitivity values independent of the reference set.
> >
> >
> >
> >
> >
> > > *Figure 3: The interpretation of the results are not clear. Do we know what the differences in the mechanisms of MDS and RTE are that leads to this inverted pattern? It would be very helpful to interpret the result if there is some brief explanation on what those methods are and why they are used.*
> >
> >
> > Whereas MDS (Kruskal, 1964) visualises similarities based on pairwise differences of data points, RTE (Geurts et al., 2006; Moosmann et al., 2006) is based on fitting a random forest to the data points, whose representations are then determined by the leaves they end up in. Naturally, nearby points are more likely to fall into the same leaf. For the details, we refer to the corresponding papers. The inverted patterns are explained by the different implementations. The critical aspect is whether points of the same colour are grouped together, which expresses similarity in both cases.
> > To clarify this aspect, we added more explanations regarding the functionality of MDS and RTE (due to space constraints, we moved the RTE plots to the Appendix).
> >
> >
> > > *Exploration experiments: In the setup column of Table2, just to clarify: minus v, "-v" means removing v from the training set, and (v) means all the validation accuracy is based on v. Is this correct?*
> >
> > Correct.
> >
> > > *Are the most sensitive data points computed based on another pre-trained model?*
> >
> > No. Sensitivity information is based entirely on the feature and label information of the data (comp. Equation 1). In other words, the most and least sensitive points are defined in a model-agnostic way, allowing us to predict which regions a model perceives as areas of high or low curvature -before- a model is trained.
> >
> >
> > ---
> > ---
> > ---
> > ## *Regarding the suggestions.*
> > ---
> >
> >
> > > *General suggestions on all figures: Please consider using subcaptions to increase the clarity of the results.*
> >
> > We restructured the graphics and divided them into sub-graphics with individual captions.
> >
> >
> >
> >
> >
> >
> >
> >
> >
> >
> >
> > ---
> > ---
> > ---
> > ## *References*
> > ---
> >
> > Zekai Wang, Tianyu Pang, Chao Du, Min Lin, Weiwei Liu, and Shuicheng Yan. Better diffusion models further improve adversarial training. In Andreas Krause, Emma Brunskill, Kyunghyun Cho, Barbara Engelhardt, Sivan Sabato, and Jonathan Scarlett (eds.), Proceedings of the 40th International Conference on Machine Learning, volume 202 of Proceedings of Machine Learning Research, pp. 36246–36263. PMLR, 23–29 Jul 2023.
> >
> > ---
> >
> > Yair Carmon, Aditi Raghunathan, Ludwig Schmidt, Percy Liang, and John Duchi. Unlabeled
> > data improves adversarial robustness. In Advances in Neural Information Processing Systems
> > (NeurIPS), 2019.
> >
> > ---
> >
> > Antonio Torralba, Rob Fergus, and William T. Freeman. 80 million tiny images: A large data set for
> > nonparametric object and scene recognition. IEEE Transactions on Pattern Analysis and Machine
> > Intelligence, 30(11):1958–1970, 2008.
> >
> > ---
> >
> > Alex Krizhevsky. Learning multiple layers of features from tiny images. 2009.
> >
> > ---
> >
> > J.B. Kruskal. Nonmetric multidimensional scaling: A numerical method. Psychometrika, 29(2):
> > 115–129, 1964.
> >
> > ---
> >
> > Pierre Geurts, Damien Ernst, and Louis Wehenkel. Extremely randomized trees. Machine Learning,
> > 63:3–42, 2006.
> >
> > ---
> >
> > Frank Moosmann, Bill Triggs, and Frederic Jurie. Fast discriminative visual codebooks
> > using randomized clustering forests. In B. Schölkopf, J. Platt, and T. Hoffman
> > (eds.), Advances in Neural Information Processing Systems, volume 19. MIT Press,
> > 2006.

---

> > > ### Comment · Reviewer_vUSY · 2023-11-22
> > >
> > > Thank you for your response. I will go through the revised paper shortly and then go through your responses.
> > > Meanwhile, I have a quick question. How statistically significant is the 0.2% improvement? Is it an average over multiple runs? Given this modest improvement, particularly, it is important to report other statistics such as the total number of runs, the standard deviation between the runs, etc.

---

> > > > ### Author Response · Authors · 2023-11-23
> > > > **Answer to Official Comment by Reviewer vUSY**
> > > >
> > > > These are one-time statistics because the training procedure is computationally expensive (2.5 days of training over four 40GB GPUs for one model alone).
> > > >
> > > > However, as shown in Table 2, the adversarial accuracy when focusing on a randomly chosen set or no set is consistent (first three rows). Analogously, we observe consistent improvements in rows 5-7, indicating that the effect of focusing on the least robust elements is consistent as well.

---

### Official Review · Reviewer_8FHc · 2023-11-03

**Soundness:** 3 good
**Presentation:** 2 fair
**Contribution:** 2 fair
**Rating:** 3
**Confidence:** 3

**Summary:**

The authors investigate the robustness of neural networks undergoing training via gradient descent through the lens of the geometry of the data. The authors analyze the dynamic of robustness by proposing a measure of “perceived curvature”. Essentially, the perceived curvature resembles the local Lipschitz constant exhibited by the neural network, modified so that the predictions are mapped to the discrete set of labels. Algorithmically, the authors analysis implies that by emphasizing the least-robust elements of the training set, modest gains in adversarial test error can be achieved.

The authors perform exploratory experiments by showing some correlation between the perceived manifold curvature and robustness as well as visualizations depicting the most and least robust examples and data sensitivity.

While the paper is interesting and the experiments are reasonably comprehensive, I do not think this paper offers particularly new or deep insight into the nature of adversarial robustness, beyond what has been explored by prior work. These issues coupled with the quality of the writing and composition make me inclined to reject, although I am open to changing my score.

**Strengths:**

- Interesting application of diffusion models to investigate the adversarial robustness of a neural network for certain examples via a notion of perceived curvature
- Comprehensive visualizations, plots to demonstrate the relationship between per-sample-robustness, sensitivity, and margin

**Weaknesses:**

**Contribution / significance**

The basic observation made by the paper regarding the relationship between robustness, sensitivity, and sample importance during training is interesting, but well-known. To strengthen the contribution and significance of the work, the authors should clarify the contribution of their analysis in the context of the previous work, or demonstrate some actionable insights- e.g. an algorithm that exhibits superior adversarial robustness relative to existing techniques.

**Missing relevant work**

There is some missing existing work that should be cited that explores the emphasis of certain vulnerable examples in the training set to enhance clean and robust test-set performance. E.g. reweighting methods such as [1, 2], subsampling methods such as [3],  and others that I do not list (e.g. on applications of diffusion models to the adverarial robustness context).

[1] Zhang et al., Geometry-aware Instance-reweighted Adversarial Training, ICLR 2021

[3] Wang et al., Probabilistic Margins for Instance Reweighting in Adversarial Training, NeurIPS 2021

[3] Zhang et al., Attacks Which Do Not Kill Training Make Adversarial Learning Stronger, ICML 2020

**Writing and composition**

The writing could use some work. Several seemingly important statements are made, but I found it difficult to parse the english. For example, the following are examples:

_However, we argue that the labels evoke the impression of disconnectedness, which a model then tries to account for when remodelling the perceived decision boundaries during training._

_Although the skewed label distribution of s (comp. Figure 5) should come as a disadvantage, one may, a priori, argue for the converse…_

I also could not understand the idea of figure 1. The preceding text states that they intend to illustrate some previous claim, but the previous claim seems to be about data sensitivity and curvature, while figure 1 details the adversarial robustness / robustness gap for models trained for different numbers of epochs with a certain overlap between the training and validation sets. The experiment seems very complicated and difficult to understand compared to the claim that

_It appears beneficial to emphasise the least robust elements during training to help a model gain robustness in regions where it struggles the most._

**Questions:**

- Could the authors clarify their contribution in the context of existing methods / analysis (e.g. by providing some explanation for the efficacy of existing methods to enhance robustness)?
- One claim is that _diffusion model connects regions of non-robust data to more prominent semantic changes, which we take as the model accounting for a more significant perceived curvature._ Can this be made more precise?

---

> ### Author Response · Authors · 2023-11-21
> **Answer to Official Review of Submission3684 by Reviewer 8FHc**
>
> ## *We thank reviewer "8FHc" for the criticism and their literature recommendations.*
> ---
> ---
> ---
> ## ---*First, we address the mentioned weaknesses individually.*---
> ---
> **Contribution / significance**
>
> To explain and delineate our contribution from existing work and to show the significance of the proposed link between data robustness and the perceived curvature of a model, we reframed our exposition in the updated version.
>
> As mentioned by the reviewer, it is known that nearby differently labelled points (= non-robust points) enforce small margins, which naturally impair adversarial training in such regions (as showcased in the first example). However, while "less space" may explain the weaker effect of adversarial training in such non-robust regions, it fails as an explanation if "less space" is "still more than enough space". To strengthen our argument, we included the following point: the minimal $l_\infty$ distance between two non-robust CIFAR-10 images ($\approx0.2118$) is more than six times as large as the maximum perturbation radius ($\approx0.0314$) which, geometrically, leaves more than enough space for a robust decision boundary. Nevertheless, a robust decision boundary is not found during training for a significant portion of the non-robust points (comp. Fig. 1). As our primary contribution in this work, we provide a new perspective for this phenomenon to explain why these non-robust points are "difficult" for any model and why adversarial training may not lead to a robust decision boundary, even if we know it exists.
>
> Whereas this motivates new adversarial training methods, the development of such a method is beyond the scope of this work. Instead, we provide theoretical and empirical evidence to show that this curvature-based perspective is meaningful in practice and can motivate new methods of adversarial training in the future. We made significant changes in the introduction and Section 4 to clarify this point.
>
> **Missing relevant work**
>
> We included the listed works to explain what phenomenon we focus on in this work (which cannot be explained by traditional margin-based optimisation strategies; see explanation above). We highlight that our primary contribution is a new and deeper geometry-based perspective to explain why adversarial training shows a weaker effect in certain regions. We clarified this point in the updated version.
>
>
> **Writing and composition**
>
> We reframed several sentences in the updated version to make the content less convoluted and clarified some essential aspects to make them more accessible. To this end, we redesigned the example in the introduction as well.
>
> ---
> ---
> ---
> ## ---*Regarding the questions.*---
> ---
> > *Could the authors clarify their contribution in the context of existing methods / analysis (e.g. by providing some explanation for the efficacy of existing methods to enhance robustness)?*
>
> Our primary contribution is an explanation for the weaker effect of adversarial training in non-robust regions, which small margins between differently labelled points cannot entirely account for. This explanation, building on connections between a model’s perception of curvature and the robustness of data, can motivate new methods of adversarial training in the future. In this work, we aim to provide theoretical and empirical evidence to demonstrate that this connection exists and is meaningful in addition to the development of new methods (which is beyond the scope of this paper).
>
>
> > *One claim is that diffusion model connects regions of non-robust data to more prominent semantic changes, which we take as the model accounting for a more significant perceived curvature. Can this be made more precise?*
>
> We rephrased the explanation in the paper to clarify what we mean. In essence, we use the diffusion model to generate new images from the most and least robust regions. As we demonstrate, the semantic differences between the original and generated images are much smaller for the most robust elements than the least robust ones. Hence, the semantic distance (or manifold distance $\tilde{d}$) is larger for the latter. By comparing the corresponding $l_2$ and $l_\infty$ distances between original images and clones, we notice that the converse holds for the pixel distance. More precisely, the pixel distance ($d$) between the original and generated images is more significant for the most robust elements compared to the least robust elements.
>
> By comparing the semantic and pixel distances in the form of the Finsler-Haantjes curvature, we see that the diffusion model connects regions of lower/higher curvature with regions of robust/non-robust data. Here, we refer to regions instead of single points, as curvature and robustness (or, reciprocally, sensitivity) are local concepts (comp. Theorem 1).

---

### Official Review · Reviewer_yQLb · 2023-11-06

**Soundness:** 2 fair
**Presentation:** 1 poor
**Contribution:** 2 fair
**Rating:** 3
**Confidence:** 3

**Summary:**

The paper is on adversarial robustness and proposes that models are particularly vulnerable in regions where the data manifold would be perceived as highly curved by the model. Some theoretical developments are proposed to support that. Experiments are conducted to demonstrate that by oversampling data samples in curved areas and using them to generate new artificial samples for training would improve robustness.

**Strengths:**

- Tackle a significant question on understanding better the input space of deep models and the corresponding robustness to adversarial attacks.
- Support claims through elaborated theoretical developments.

**Weaknesses:**

- The paper and overall presentation is very difficult to follow. Although the authors seem to know very well their topic, the communication is lacking and a reader not in that specific field gets lost quite quickly.
- The notion of curvature on the manifold is really unclear to me and not very well explained in the paper. But it appears in the end we are looking at distance between samples, the notion of curvature is there to support theoretical developments that are not directly translated in practice.
- The technical aspects of the experiments section are not very clear nor clearly explained. I guess that the reader should look at some of the referenced papers like Karras et al. (2022) and Wang et al. (2023), but still I would like to get more background and specific details to better understand what is done in the experiments. It is quite unclear to me that the details provided would make results reproducibility easy.
- It is difficult to figure out what exactly the experimental results are providing as support to the conclusion. The differences in Table 2 between the results is very small, and as such not very convincing that the proposal is of any meaningful effects for improving robustness.
- Overall, the experiments are not very well explained and presented and the results are very difficult to interpret. I have a hard time making sense of all this.

**Questions:**

Looking at equation 1, if we assume that $d(p_i,p_j)$ is an Euclidean distance and that $\|y(p_i)-y(p_j)\|$ is basically equal to zero or one when using $y(p)$ as one hot vector over the classes, it means that in practice, the proposal consists of looking 1/distance to the nearest sample from a different class from the current one. Is this correct? That’s what was used for the experiments?

**Details Of Ethics Concerns:**

No ethics concerns with this paper.

---

> ### Author Response · Authors · 2023-11-21
> **Answer to Official Review of Submission3684 by Reviewer yQLb**
>
> ## *We thank reviewer "yQLb" for the criticism.*
>
> ---
> ---
> ---
>
> ## ---*First, we address the mentioned weaknesses regarding the exposition and clarity of our paper.*---
> ---
> In order to make the content of our paper more accessible, we have rewritten the introduction. The primary goal was (i) to make abstract concepts such as data robustness and the data manifold's curvature more accessible and (ii) to clarify what our contribution in this context is. We removed two plots in Figure 1 and added a schematic to clarify what we mean by the different distances in feature space and along the data manifold (as a preliminary for curvature).
>
> To show why perceived curvature is meaningful, we outlined how it can explain the weaker performance of adversarial training in regions of non-robust data where robust decision boundaries exist but are not found during training.
>
> To clarify how our experimental evidence supports our theoretic view (of a model perceiving curvature differently in robust and non-robust regions), we have also rewritten Section 4. One goal was to convey an intuition for why we observe a diffusion model generating semantically very different images in the least robust regions but not in the most robust regions. Overall, we explained the experimental setups in a less convoluted way to avoid confusion.
>
> Finally, the gains in adversarial robustness appear small but are indeed significant in this area, where we refer to the work of **Wang et al. (2023)**. However, the surprising fact we stressed in the updated version is that focusing on a negligible number of points from non-robust regions leads to an increased robust generalisation accuracy beyond the expected local improvements. In other words, a better understanding of the pixel distances between points is paired with a better understanding of semantic distances (hence the improved performance on unseen data).
>
> ---
> ---
> ---
>
> ## ---*Regarding the question.*---
> ---
> Yes. This “sensitivity” information was used in the experiments. However, as shown theoretically, we are not interested in the absolute sensitivity value but the value compared to the entire sensitivity spectrum. In this way, sensitivity has only meaning in relative terms. This is why we always refer to the “most and least sensitive or robust points” instead of the “points with a sensitivity above/below x”. Ultimately, this allows us to connect the primitive measure of sensitivity to the elaborate concept of curvature.
>
> ---
> ---
> ---
>
> ## ---*References*---
> ---
> Zekai Wang, Tianyu Pang, Chao Du, Min Lin, Weiwei Liu, and Shuicheng Yan. Better diffusion models
> further improve adversarial training. In Andreas Krause, Emma Brunskill, Kyunghyun Cho, Barbara
> Engelhardt, Sivan Sabato, and Jonathan Scarlett (eds.), Proceedings of the 40th International
> Conference on Machine Learning, volume 202 of Proceedings of Machine Learning Research, pp.
> 36246–36263. PMLR, 23–29 Jul 2023.

---

### Author Response · Authors · 2023-11-21
**General statement regarding our submission.**

We first thank all reviewers for their criticism and recommendations regarding our paper, its exposition and clarity.

Given the comments and criticism, we revised large portions of our exposition to clarify our contribution and how our theoretical and empirical results support our proposed view. The revised version follows a different presentation style to make abstract concepts more accessible to readers and provide better arguments on why our paper adds a meaningful new perspective to the field of representation learning. Below, we list a change log. The reviews will be addressed individually.


(i) We rewrote the abstract and the introduction to point out the problem in adversarial training we address and provide a new explanation for, encapsulating our primary contribution.

(ii) We modified the first example and substituted two plots to explain concepts such as robustness of data and (perceived) curvature in a more accessible way.

(iii) We added minor changes to the related work section and the theoretical background.

(iv) We restructured portions of Section 4 to clarify how our experimental results support our theoretical claims in a more accessible and less convoluted style.


-The authors-

---

### Meta-Review · Area_Chair_VVZm · 2023-12-07

**Metareview:**

The paper proposes that models exhibit stronger vulnerability in regions where the data manifold appears highly curved to the model. The authors propose theoretical developments to support this claim and conduct experiments demonstrating that oversampling data from curved areas and utilizing them to generate new training samples can enhance robustness.

Strength:
This study offers an interesting and significant perspective on the subject matter. The comprehensive visualizations aid in a better understanding of the concepts presented.

Weaknesses:
As highlighted by the reviewers, the work lacks novelty and overlooks important related research. Additionally, multiple reviewers expressed concerns about the paper's organization and writing quality.

Hence, all reviewers agree that this work requires substantial improvement, and the current version does not meet the acceptance criteria for ICLR.

**Justification For Why Not Higher Score:**

As highlighted by the reviewers, this work lacks novelty and misses important related works. Furthermore, multiple reviewers expressed concerns about the paper's poor organization and writing. Consequently, we recommend rejecting this submission.

**Justification For Why Not Lower Score:**

N/A

---

### Decision · Program_Chairs · 2024-01-16

Reject